# JacobiGAD: Jacobi Polynomial–Powered Heterogeneous Graph-Level Anomaly Detection

## Abstract

Heterogeneous graph-level anomaly detection is vital for applications such as fraud detection and drug discovery, yet remains challenging due to mixed features, complex structures, and severe class imbalance. This paper introduces JacobiGAD, a unified framework that addresses these challenges through three key innovations. First, learnable multiscale filters based on Jacobi Polynomials adapt to different node and edge types, fusing multiple graph views to enhance anomaly signals. Second, these polynomials enable efficient approximation of targeted functions and naturally encode diverse geometries. Third, a Ricci Flow-inspired loss amplifies gradients for rare anomalies, mitigating class imbalance without distorting graph embeddings, ensuring stable convergence. Extensive experiments on real-world benchmarks show JacobiGAD outperforms the best baseline by up to 2.79% (AUROC), 7.78% (AUPRC), 7.11% (Recall@k), and 5.96% (F1-score) on average.

## 1 Introduction

Graph-level anomaly detection (GAD) identifies entire graphs that exhibit structural or attributively deviations from norm ones, a critical task for applications (Ma et al., 2023; Lin et al., 2024), such as financial fraud detection, drug toxicity screening, and infrastructure monitoring. These graphs are often heterogeneous, containing multiple node and edge types, presenting three core challenges: Mixed feature spaces (Xu et al., 2024). Heterogeneous graphs combine diverse attributes with varying dimensions, making it hard to design unified filters that capture relevant anomalous patterns across all types. (2) Structural complexity (Zhang et al., 2022). Multiple edge types and intricate structures create difficulties in detecting anomalous substructures within graphs, which can easily mislead the detector. (3) Imbalanced label distribution (Dong et al., 2024). Genuine anomalies are exceedingly rare, leading to highly skewed training sets that bias models toward normal graphs. Existing methods for graph classification or GAD struggle to surface anomalous signals in such imbalanced heterogeneous data, as illustrated in Section 2 and demonstrated in Section 5.

To tackle these challenges, this paper presents JacobiGAD, an end-to-end framework that unifies adaptive multi-scale spectral filters with imbalance-sensitive loss for heterogeneous GAD. Specifically, our learnable Jacobi Polynomial filters adapt to diverse node and edge types across multiple graph views, enhancing true anomaly signals while suppressing noise. A complementary Ricci Flow-inspired loss dynamically amplifies gradients for rare anomalies, effectively combating class imbalance. Theoretically, we prove that these filters enable fast, stable approximation while preserving feature distances, and that the loss ensures reliable convergence. Empirically, JacobiGAD consistently outperforms all baselines across AUROC, AUPRC, Recall@k, and F1-score on 15 real-world benchmarks. In summary, our contributions are threefold:

- We propose JacobiGAD, a novel framework for heterogeneous GAD that integrates adaptive Jacobi Polynomial filters with a Ricci Flow-inspired loss function.

- We provide theoretical guarantees on filter stability, information preservation, and loss convergence, ensuring principled and efficient learning.

- We comprehensively validate JacobiGAD on diverse real-world datasets, showcasing its superior ability to detect rare anomalies that existing methods fail to identify.

## 2 RELATED WORK

**Homogeneous Graph Classification.** Early successes in graph classification on homogeneous networks include GCN (Kipf & Welling, 2017), which approximates spectral graph convolutions, SAGE (Hamilton et al., 2017), which samples neighborhoods, GAT (Velickovic et al., 2018), which applies attention to neighbor messages, and GIN (Xu et al., 2019), which demonstrated that sum-aggregation matches the Weisfeiler–Leman test's expressivity. Recent extensions such as LRGNN (Wei et al., 2023) stack GNNs for long-range dependencies, GRDL (Wang & Fan, 2024) treats node embeddings as discrete distributions for direct classification, UQGNN (Wu et al., 2025) introduces uncertainty-aware objectives for robustness, and UIL (Sui et al., 2025) offers a unified view on invariant graph learning. While effective on balanced, homogeneous benchmarks, these models struggle with integrating multiple node/edge types and detecting rare anomalies in complex structures.

**Heterogeneous Graph Classification.** Methods such as HMGNN (Yu & Gao, 2022) and muxGNN (Melton & Krishnan, 2023) capture heterogeneity using motifs or multiplex networks, while HeGCL (Shi et al., 2024) employs contrastive learning on multiple views. Subsequent approaches, such as RFAGNN (Wu et al., 2024) and SHGLNN (Hayat et al., 2024), use relational attention or hypergraphs to model complex interactions. Although these methods perform heterogeneous graph classification, they rely on fixed filters or heuristic fusion strategies, assume balanced data, and lack principled mechanisms for anomaly detection.

**Graph-level Anomaly Detection.** Current anomaly detection literature includes iGAD (Zhang et al., 2022), which learns anomalous substructures in graphs, GmapAD (Ma et al., 2023), which maps graphs into feature spaces based on similarity to representative nodes, RumorMixer (Xu et al., 2024), focusing on the echo chamber effect and platform heterogeneity; RQGNN (Dong et al., 2024), which uses the Rayleigh Quotient to uncover sample properties, and UniGAD (Lin et al., 2024), which tackles multi-level tasks for diverse information. While these methods perform well in GAD, they struggle to generalize to heterogeneous scenarios due to their inability to adapt filters to diverse feature domains, fuse multiple graph views, and incorporate theoretically guaranteed loss for handling imbalanced data in complex structures.

In contrast, JacobiGAD is an innovative end-to-end framework specifically designed for heterogeneous GAD. It introduces learnable, multi-scale spectral filters that adaptively fuse signals across diverse node and edge types, and a Ricci Flow–inspired loss that counteracts class imbalance by dynamically emphasizing rare anomalies. Unlike homogeneous methods, it natively handles heterogeneous complexity; unlike existing heterogeneous classifiers, it uses learned, geometry-aware filters instead of fixed bases; and unlike all prior approaches, it addresses severe imbalance in a theoretically grounded manner, enabling the detection of subtle anomalies that are missed by other methods.

## 3 PRELIMINARIES

**Heterogeneous Graph.** A heterogeneous graph is defined as $G = (\mathcal{V}, \mathcal{A}, \mathcal{X}, T_V, R_E)$, where the node set $\mathcal{V} = \bigcup_{t=1}^{|T_V|} \{V_t\}$ comprises $|T_V|$ distinct types of nodes, each endowed with an attribute matrix $\boldsymbol{X}_t \in \mathbb{R}^{|V_t| \times d_t}$ in $\mathcal{X}$. The set of adjacency matrices $\mathcal{A} = \{\boldsymbol{A}_r\}_{r=1}^{|R_E|}$ with each $\boldsymbol{A}_r \in \mathbb{R}^{|\mathcal{V}| \times |\mathcal{V}|}$, encodes the $|R_E|$ relation types by setting $(\boldsymbol{A}_r)_{ij} = 1$ if nodes $i, j \in \mathcal{V}$ are linked under relation $r$, otherwise $(\boldsymbol{A}_r)_{ij} = 0$. The schema is completed by $T_V$, the set of node types, and $R_E$, the set of relation types, which together satisfy $|T_V| + |R_E| > 2$. In practice, heterogeneous graphs often exhibit heterogeneity in attribute dimensions, i.e., $d_{T_i} \neq d_{T_j}$ for $T_i \neq T_j$, $T_i, T_j \in T_V$.

**Task Definition.** Given a heterogeneous graph set $\mathcal{G} = \{G^{(i)} = (\mathcal{V}^{(i)}, \mathcal{A}^{(i)}, \mathcal{X}^{(i)}, T_V, R_E)\}_{i=1}^{N}$, we partition $\mathcal{G}$ into two disjoint subsets, anomalous graphs $\mathcal{G}^{an}$ and normal graphs $\mathcal{G}^{no}$, with $\mathcal{G}^{an} \cap \mathcal{G}^{no} = \emptyset$. The GAD task then seeks to assign each $G^{(i)} \in \mathcal{G}$ to one of these classes, based on atypical structural or attribute patterns that distinguish anomalous instances. Beyond the difficulties of complex feature and structure caused by heterogeneity, heterogeneous GAD also exhibits severe class imbalance, i.e., $|\mathcal{G}^{an}| \ll |\mathcal{G}^{no}|$, which compounds the difficulty of reliable anomaly discrimination. Building on this formulation. Our study proposes a novel spectral GNN based on Jacobi Polynomials under the guidance of Ricci Flow-inspired loss, specifically designed for heterogeneous GAD to address the challenges mentioned in Section 1.

**Spectral Graph Neural Network.** The key ideas of spectral GNNs are to conduct graph convolutional operations in the Fourier domain, which can be defined as $g \star \boldsymbol{X} = g(\boldsymbol{L})\boldsymbol{X}$, where $g(\cdot)$ is the graph filter, $\boldsymbol{X}$ is the feature matrix of the graph, and $\boldsymbol{L}$ is the normalized Laplacian matrix, which can be defined as $\boldsymbol{L} = \boldsymbol{I} - \boldsymbol{D}^{-\frac{1}{2}} \boldsymbol{A} \boldsymbol{D}^{-\frac{1}{2}}$, given the adjacency matrix $\boldsymbol{A}$, corresponding degree matrix $\boldsymbol{D}$, and an identity matrix $\boldsymbol{I}$. The successful choices of $g(\cdot)$ from prior work (Defferrard et al., 2016), are polynomials, inspiring our exploration of the optimal basis of the graph filter in heterogeneous GAD.

**Jacobi Polynomials.** Jacobi Polynomials $\{P_n^{\alpha,\beta}(x)\}_{n=0}^{\infty}$ are a family of orthogonal polynomials on the interval $x \in [-1, 1]$ with weight function $w(x) = (1-x)^{\alpha}(1+x)^{\beta}$ for parameters $\alpha, \beta > -1$:

$$P_0^{\alpha,\beta}(x) = 1,$$

$$P_1^{\alpha,\beta}(x) = (\alpha + 1) + \frac{\alpha + \beta + 2}{2}(x - 1),$$

$$P_k^{\alpha,\beta}(x) = (\theta_k^{(1)} x + \theta_k^{(2)}) P_{k-1}^{\alpha,\beta}(x) - \theta_k^{(3)} P_{k-2}^{\alpha,\beta},$$

where

$$\theta_k^{(1)} = \frac{(2k + \alpha + \beta - 1)(2k + \alpha + \beta)}{2k(k + \alpha + \beta)},$$

$$\theta_k^{(2)} = \frac{(2k + \alpha + \beta - 1)(\alpha^2 - \beta^2)}{2k(k + \alpha + \beta)(2k + \alpha + \beta - 2)},$$

$$\theta_k^{(3)} = \frac{(k + \alpha - 1)(k + \beta - 1)(2k + \alpha + \beta)}{k(k + \alpha + \beta)(2k + \alpha + \beta - 2)}$$

Jacobi Polynomials provide a general solution for graph signal filtering. In more detail, increasing $\alpha$ decreases contributions near the upper end of the spectrum, i.e., high-frequency or rapidly varying components, while increasing $\beta$ down-weights contributions near the lower end, i.e,. low-frequency or smooth components. In practice, this parametrization yields an efficient, $k$-hop localized graph convolution operator whose passband can be finely tuned by selecting $\alpha$ and $\beta$ to match the topology and signal characteristics of diverse graph domains. Special cases include classical polynomials, such as Legendre Polynomials ($\alpha = \beta = 0$), Chebyshev Polynomials ($\alpha = \beta = -\frac{1}{2}$), and Gegenbauer Polynomials ($\alpha = \beta = \lambda - \frac{1}{2}$, where $\lambda > -\frac{1}{2}$).

**Ricci Flow.** Ricci Flow is a geometric process that deforms a Riemannian metric $g(t)$ according to:

$$\frac{\partial g(t)}{\partial t} = -\gamma \text{Ric}(g(t)),$$

where $\text{Ric}(g(t))$ denotes the Ricci curvature tensor and $\gamma \in \mathbb{R}^+$. Under this evolution, regions of high curvature flatten out, leading to a more uniform geometry. In graphs, edgewise curvature measures are defined via optimal transport between local neighborhood distributions. A discrete Ricci Flow then updates edge weights to equalize the curvature across the graph. This curvature-guided objective counteracts extreme class imbalance without globally distorting the graph representation, ensuring that rare but structurally distinctive anomalies receive proportionally larger gradient updates.

## 4 METHOD

### 4.1 OVERVIEW

In this section, we present an overview of JacobiGAD in Figure 1. First, we unify heterogeneous features via Gaussian projection and construct a multi-view topology in Section 4.2. Additionally, we demonstrate the distance preservation property of our alignment, as shown in Theorem 1. Next, we propose JPGNN, a spectral filter based on Jacobi Polynomials, which fuses multiple views while provably preserving feature and structural information in Section 4.3. Furthermore, we prove the optimal basis, information preservation, target amplification, multiple spaces, extensive approximation, and converged approximation properties of JPGNN, as shown in Theorems 2, 3, 4, 5, 6, and 7, respectively. Finally, we introduce RFACE, a Ricci Flow-inspired loss, that intrinsically adapts to imbalanced distributions in Section 4.4. Morevoer, we verify the weight balance and convergence guarantee properties of RFACE, as shown in Theorems 8 and 9.

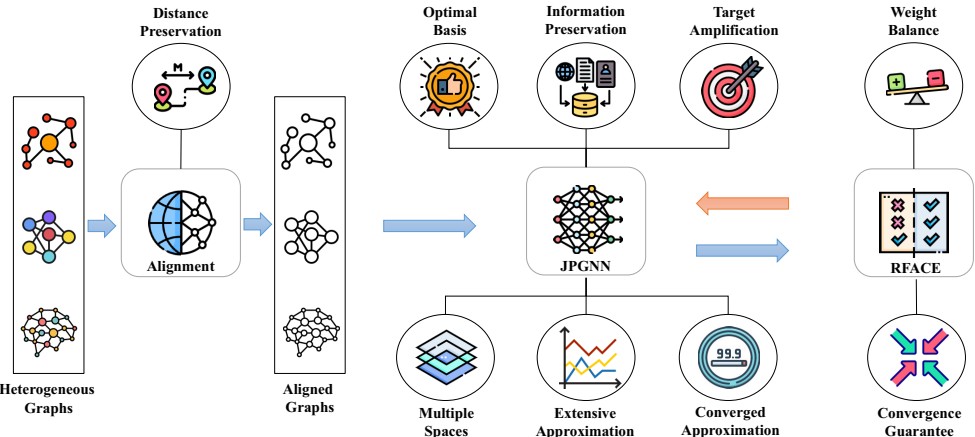

Figure 1: Overview of JacobiGAD.

## 4.2 Heterogeneous Feature Projection and Multi-View Topology

The initial processing of heterogeneous data is a critical determinant of model performance. Conventional approaches often fail to adequately address two fundamental challenges: the misalignment of features across node types and the synergistic integration of multiple relational contexts. Our method addresses these challenges through a principled framework.

**Heterogeneous feature projection.** Current methods for handling heterogeneous node features, such as decomposition (Ren & Du, 2020), concatenation (Gao et al., 2023), and independent learning (Yang et al., 2023), each face significant limitations. Decomposition reduces dimensionality at the cost of information loss. Concatenation increases feature size, leading to overfitting and high computational cost. Independent learning ignores semantic alignment between views, introducing noise and raising training expense. All fail to adequately align semantics across node types.

To address the above drawbacks, we propose a theoretically guaranteed approach that can align features without severe information loss and requires no parameter tuning. Given a set of heterogeneous graphs with in total $|T_V|$ types of node features, we first apply zero padding to the end of each type so that all types have the same dimensionality $d_{max}$, i.e., $\boldsymbol{X}_t' = \boldsymbol{X}_t \oplus \boldsymbol{0}$, where $\boldsymbol{X}_t' \in \mathbb{R}^{|V_t| \times d_{max}}$ and $\boldsymbol{0} \in \mathbb{R}^{|V_t| \times (d_{max}-d_t)}$, $\forall t \in T_V$. Then, we use a shared Gaussian matrix to project them into a lower latent space with dimension $d$, i.e., $(\boldsymbol{X}_t^{proj})^T = \boldsymbol{P}(\boldsymbol{X}_t')^T$, where $\boldsymbol{P} \in \mathbb{R}^{d \times d_{\max}}$ and $\boldsymbol{P}_{i,j} \sim \mathcal{N}(0, \frac{1}{d})$. The result feature matrix $\boldsymbol{X}^{proj} = \boldsymbol{X}_1^{proj} \oplus \boldsymbol{X}_2^{proj} \oplus \cdots \oplus \boldsymbol{X}_{|T_V|}^{proj}$. The validity of this approach is rigorously guaranteed by the following theorem, whose proof is in Appendix A.

**Theorem 1.** *Given any finite set of vectors with different dimensions, zero-padding at any position can equivalently preserve their original information. For any zero-padded vector $\boldsymbol{x} \in \mathbb{R}^D$, a data-independent Gaussian projection $f(\boldsymbol{x}) = \boldsymbol{P}\boldsymbol{x}$, where $\boldsymbol{P} \in \mathbb{R}^{d \times D}$ and $\boldsymbol{P}_{i,j} \sim \mathcal{N}(0, \frac{1}{d})$, can preserve pairwise Euclidean distance for $M$ pairs up to a factor $\epsilon$ with high probability, $1 - 2Me^{-\frac{\epsilon^2 d}{4}}$.*

Theorem 1 shows that high-dimensional vectors can be projected into a lower-dimensional space while preserving their pairwise distances with high probability. This ensures the semantic relationships between nodes are maintained isotropically in the latent space, without additional cost during training.

**Multi-view topology.** Additionally, to address the multifaceted topology of heterogeneous graphs, we move beyond the naive summation of adjacency matrices, which assumes all relation types are equally important. Instead, we employ learnable weights for each relation: $\tilde{\mathbf{A}} = \sum_{r=1}^{|R_E|} \omega_r \mathbf{A}_r$. This allows the model to dynamically discern the hierarchical importance of different relational contexts. However, prior heuristic weighting schemes often fail to leverage inherent structural patterns. Our method, detailed in Section 4.3, provides a theoretically grounded approach for optimal multi-view fusion, ensuring convergence and information preservation to leverage the formulation effectively.

### 4.3 Jacobi Polynomial-based Graph Neural Network

Our spectral GNN takes as input a graph with aligned features and fused topology. Selecting an appropriate spectral filter basis is critical for heterogeneous GAD, as different bases offer distinct expressivity. We posit that Jacobi Polynomials are the optimal basis, a claim supported by the following analysis and theorems. First, we consider homogeneous graph classification, a closely related case of heterogeneous GAD. The core of our argument rests on a theorem, proved in Appendix A, establishing the direct and general optimality of Jacobi Polynomials.

**Theorem 2.** *Consider the optimization process of a spectral GNN in graph classification: $argmin_{\theta_k, \boldsymbol{W}} \mathcal{L}(\boldsymbol{y}, Pooling(\sum_{k=1}^{K} \theta_k g_k(\boldsymbol{L}) \boldsymbol{X} \boldsymbol{W}))$. Assuming that it can reach a global minimum by tuning learnable parameters $\theta_k, \boldsymbol{W}$, then the optimal choice of the basis of the graph filter $g_k(\cdot)$ can be the Jacobi Polynomials, according to its convergence speed to the minimum area.*

Theorem 2 provides the foundational justification for our architecture, demonstrating that Jacobi Polynomials are optimal for the case of homogeneous graph classification. This inspires their use for more complex heterogeneous graph-level tasks. As established in Section 4.2, a heterogeneous graph's multi-view topology is a linear combination of homogeneous adjacency matrices. Consequently, the task can be represented as a combination of its homogeneous variants. Therefore, the expressivity of Jacobi Polynomials for heterogeneous GAD hinges on their ability to filter and fuse this multi-view information, a capability demonstrated by the following theorems, whose proofs are in Appendix A.

**Theorem 3.** *Given different views of a graph, the combination of Jacobi Polynomial-based graph filter can preserve the full information from the original graph due to injectivity.*

**Theorem 4.** *Combining information from $V$ views using the combination of Jacobi Polynomial-based graph filter will amplify targeted patterns (the enhancement factor grows as $\Theta(V)$) while suppressing noise (the signal-to-noise ratio grows as $\Theta(\sqrt{V})$).*

Theorem 3 and 4 demonstrate that a Jacobi Polynomial-based filter comprehensively preserves information while selectively amplifying targeted patterns and reducing noise. This is vital for heterogeneous GAD, where anomalies are often subtle inconsistencies across relational views. Unlike filters that may smooth over these faint cues, our Jacobi basis can be tuned to amplify cross-view discrepancies while dampening common normal signals.

However, a filter constrained to a Euclidean prior is insufficient, as anomalies can exhibit complex structures such as hierarchical or cyclical patterns (Dong et al., 2025). Effective heterogeneous GAD thus requires a filter capable of leveraging multi-geometric information from Hyperbolic (for hierarchical data) and Spherical (for cyclical data) spaces. The following theorem, proved in Appendix A, establishes that Jacobi Polynomials possess this essential capability.

**Theorem 5.** *After appropriate coordinate transformations, Jacobi Polynomials can serve as eigenfunctions of the Laplace-Beltrami operator in the $\kappa$-stereographic model (Bachmann et al., 2020). The connections for each geometry are as follows:*

- *Spherical geometry ($\kappa > 0$): The Laplace-Beltrami operator in stereographic coordinates has eigenfunctions with radial and angular parts. The angular part is handled by spherical harmonics, while the radial part satisfies a differential equation solvable by Jacobi Polynomials.*

- *Hyperbolic geometry ($\kappa < 0$): The spectrum of the Laplace-Beltrami operator is continuous, and the radial eigenfunctions are not polynomials but can be expressed as Jacobi Polynomials.*

- *Euclidean geometry ($\kappa = 0$): The Laplace-Beltrami operator reduces to the standard Laplacian, and the radial eigenfunctions are Bessel functions, which arise as a limit of Jacobi Polynomials.*

Theorem 5 elevates our model beyond Euclidean-centric approaches. By adjusting its parameters $(\alpha, \beta)$, the Jacobi filter performs a soft selection of the optimal geometric domain for the fused graph's structure. This enables a single model to detect anomalies manifesting in any of these paradigms, a critical capability for complex real-world heterogeneous graphs.

In summary, Jacobi Polynomials offer key advantages for our task: performance guarantee, effective multi-view fusion, and adaptability to complex structural patterns. This naturally raises the question of whether a Jacobi Polynomial-based GNN can converge efficiently during training. We address this with the following theorems, which are demonstrated in Appendix A.

**Theorem 6.** *Assuming using Jacobi Polynomials as graph filter $g(\cdot)$, and the eigenvalues of the shifted Laplacian matrix $\boldsymbol{L}$ fall in $[-1, 1]$, then $g(\boldsymbol{L})$ can approximate any continuous function lying in the space $C[-1, 1]$ (contains continuous functions on $[-1, 1]$). Moreover, it can also approximate any function in the $L_w^2[-1, 1]$ space (contains measurable functions satisfying $\int_{-1}^{1} |f(x)|^2 w(x)dx < \infty$, where $w(x) = (1-x)^\alpha (1+x)^\beta$, and $\alpha, \beta > -1$).*

**Theorem 7.** *Jacobi Polynomials satisfy sharp approximation bounds. In particular, if the function $f(x)$ has $r$ continuous derivatives, then there exists a constant $C$ depending on $r, \alpha, \beta$ such that the Jacobi Polynomials $g(x)$ obeys $\min_{\deg(g(x)) \leq N} ||f(x) - g(x)||_\infty \leq \frac{C}{N} ||f^{(r)}(x)||_{L_w^1}$, which guarantees that a low-degree Jacobi filter will approximate $f(x)$ well.*

Theorems 6 and 7 guarantee our model's high expressiveness and computational efficiency. A low-order polynomial suffices to capture complex patterns, enabling a shallow architecture that avoids the over-smoothing typical of deep GNNs, a critical advantage for preserving the fine-grained differences between normal and anomalous graphs in heterogeneous GAD.

Based on Theorems 2–7, Jacobi Polynomials are theoretically justified as an optimal basis for heterogeneous GAD. We therefore operationalize this framework into a neural network layer, adhering to the parameter constraints specified in Theorem 6. We first transform the input adjacency matrix $\tilde{\boldsymbol{A}} = \sum_{r=1}^{|R_E|} \omega_r \boldsymbol{A}_r$ to the normalized Laplacian matrix $\tilde{\boldsymbol{L}} = \boldsymbol{I} - \boldsymbol{D}^{-\frac{1}{2}} \tilde{\boldsymbol{A}} \boldsymbol{D}^{-\frac{1}{2}}$, and rescale the normalized Laplacian matrix $\hat{\boldsymbol{L}} = \frac{2}{\lambda_{\max}} \tilde{\boldsymbol{L}} - \boldsymbol{I}$, where $\lambda_{\max}$ is the largest eigenvalue of $\tilde{\boldsymbol{L}}$. Then the $k$-th layer of the Jacobi Polynomial-based Graph Neural Network (JPGNN) can be defined as:

$$\boldsymbol{H}^{(k)} = \sigma((\sum_{t=0}^{T} \theta_t^{(k)} P_t^{(\alpha^{(k)}, \beta^{(k)})}(\hat{\boldsymbol{L}})) \boldsymbol{H}^{(k-1)} \boldsymbol{W}^{(k)}),$$

where $\boldsymbol{H}^{(0)} = \sigma(\boldsymbol{X}^{proj} \boldsymbol{W}^{(0)})$, $\sigma$ is a activation function, and $\omega_r, \alpha^{(k)}, \beta^{(k)}, \theta_t^{(k)}, \boldsymbol{W}^{(k)}$ are learnable parameters. Then, the graph embedding $\boldsymbol{z}$ can be obtained by:

$$\boldsymbol{H}^{stack} = \boldsymbol{H}^{(0)} \oplus \boldsymbol{H}^{(1)} \oplus \cdots \oplus \boldsymbol{H}^{(K)},$$

$$\boldsymbol{H} = \sigma(\boldsymbol{H}^{stack} \boldsymbol{W}),$$

$$\boldsymbol{z} = \text{Pooling}(\boldsymbol{H})$$

where $\boldsymbol{W}$ is a learnable parameter. This design yields a fully co-adaptive model: the multi-view fusion, governed by view weights $\omega_r$, and the spectral processing via JPGNN are jointly optimized to excel at heterogeneous GAD.

### 4.4 RICCI FLOW-INSPIRED LOSS FUNCTION

The above design addresses the first two challenges outlined in Section 1, while the final component of our framework tackles the severe class imbalance in heterogeneous GAD, where normal graphs significantly outnumber anomalies. A standard Cross-Entropy loss is ill-suited for this scenario, as it can become dominated by the majority class. To counteract this, we introduce the Ricci Flow Adjusted Cross-Entropy Loss (RFACE), which dynamically reshapes the learning landscape based on the model's output geometry.

For a graph-level classification task with $C$ classes, given predicted probability of $i$-th sample $\boldsymbol{p}_i = \text{Sigmoid}(\boldsymbol{z}_i)$, the Cross-Entropy loss is:

$$\mathcal{L}_{CE} = -\frac{1}{N} \sum_{i=1}^{N} \sum_{c=1}^{C} \boldsymbol{y}_{i,c} \log(\boldsymbol{p}_{i,c})$$

In highly imbalanced settings (e.g., $C = 2$), the standard Cross-Entropy loss, $\mathcal{L}_{CE}$, produces much larger gradients for the frequent class. This biases model updates toward the majority class, often harming minority class performance. To counteract this, we adapt principles from differential geometry, mimicking the Ricci Flow, which homogenizes a manifold's curvature. We apply this concept to the loss landscape's *curvature* per class, defined for a class $c$ as:

$$\kappa_c = \log(\frac{f_c}{\max_{c'} f_{c'} + \epsilon}),$$

Table 1: Average performance with multiple runs (homogeneous graph classification models).

| Datasets | Metrics | GCN | SAGE | GAT | GIN | LRGNN | GRDL | UQGNN | UIL | JacobiGAD |
|---|---|---|---|---|---|---|---|---|---|---|
| SF-295 | AUROC | 0.6687 | 0.7178 | 0.7409 | 0.6914 | 0.7578 | 0.6389 | 0.5248 | 0.7334 | **0.7729** |
| | AUPRC | 0.0856 | 0.1600 | 0.1645 | 0.0961 | 0.1962 | 0.0871 | 0.0525 | 0.1598 | **0.2623** |
| | Recall@k | 0.1078 | 0.2362 | 0.2099 | 0.1029 | 0.2494 | 0.1342 | 0.0519 | 0.1975 | **0.3210** |
| | F1-score | 0.4870 | 0.5685 | 0.5560 | 0.5255 | 0.4952 | 0.4871 | 0.4871 | 0.5223 | **0.6356** |
| SN12C | AUROC | 0.7034 | 0.7440 | 0.7475 | 0.7194 | 0.7747 | 0.5639 | 0.4922 | 0.7604 | **0.7797** |
| | AUPRC | 0.1032 | 0.1598 | 0.1772 | 0.1122 | 0.2038 | 0.0792 | 0.0486 | 0.1789 | **0.2666** |
| | Recall@k | 0.1279 | 0.2430 | 0.2234 | 0.1364 | 0.2515 | 0.1168 | 0.0537 | 0.2702 | **0.3240** |
| | F1-score | 0.4905 | 0.5846 | 0.5791 | 0.5242 | 0.5001 | 0.4875 | 0.4875 | 0.5105 | **0.6329** |
| UACC257 | AUROC | 0.6654 | 0.7324 | 0.6998 | 0.6848 | 0.7220 | 0.5711 | 0.4611 | 0.7124 | **0.7613** |
| | AUPRC | 0.0726 | 0.1588 | 0.1228 | 0.0903 | 0.1626 | 0.0916 | 0.0378 | 0.1134 | **0.1995** |
| | Recall@k | 0.0832 | 0.2404 | 0.1633 | 0.1136 | 0.2312 | 0.1471 | 0.0355 | 0.1562 | **0.2819** |
| | F1-score | 0.4921 | 0.5722 | 0.5214 | 0.5096 | 0.4905 | 0.4895 | 0.4895 | 0.4955 | **0.6246** |
| DBLP | AUROC | 0.9816 | 0.9746 | 0.9671 | 0.9730 | 0.7475 | 0.9800 | 0.9352 | 0.9698 | **0.9830** |
| | AUPRC | 0.9829 | 0.9758 | 0.9515 | 0.9740 | 0.6285 | 0.9761 | 0.9497 | 0.9701 | **0.9842** |
| | Recall@k | 0.9418 | 0.9441 | 0.9172 | 0.9284 | 0.6197 | 0.9374 | 0.8926 | 0.9172 | **0.9575** |
| | F1-score | 0.9598 | 0.9594 | 0.9346 | 0.9445 | 0.4863 | 0.9492 | 0.9268 | 0.3959 | **0.9651** |
| IMDB | AUROC | 0.6601 | 0.6771 | 0.6707 | 0.6601 | 0.6643 | 0.6677 | 0.6575 | 0.6487 | **0.7263** |
| | AUPRC | 0.7007 | 0.7260 | 0.7161 | 0.6948 | 0.7063 | 0.6878 | 0.6982 | 0.6764 | **0.7619** |
| | Recall@k | 0.7056 | 0.6982 | 0.6824 | 0.6982 | 0.6772 | 0.6909 | 0.6993 | 0.7003 | **0.7192** |
| | F1-score | 0.6387 | 0.6225 | 0.6045 | 0.6363 | 0.6242 | 0.6258 | 0.6305 | 0.6100 | **0.6585** |
| PDNS | AUROC | 0.7773 | 0.8577 | 0.6735 | 0.6249 | 0.8159 | 0.6935 | 0.4377 | 0.5683 | **0.8728** |
| | AUPRC | 0.4434 | 0.6110 | 0.3263 | 0.2349 | 0.5188 | 0.3578 | 0.1565 | 0.2224 | **0.6871** |
| | Recall@k | 0.4788 | 0.5900 | 0.3444 | 0.2766 | 0.5206 | 0.3299 | 0.1429 | 0.2205 | **0.6283** |
| | F1-score | 0.6743 | 0.7561 | 0.4917 | 0.4526 | 0.5718 | 0.4553 | 0.4526 | 0.4526 | **0.7760** |
| RCDD | AUROC | 0.9581 | 0.9811 | 0.9602 | 0.9658 | 0.9805 | 0.9609 | 0.8033 | 0.9593 | **0.9826** |
| | AUPRC | 0.8619 | 0.9291 | 0.8871 | 0.8823 | 0.9267 | 0.8605 | 0.3491 | 0.8645 | **0.9332** |
| | Recall@k | 0.8006 | 0.8695 | 0.8261 | 0.8249 | 0.8743 | 0.8111 | 0.4106 | 0.8081 | **0.8747** |
| | F1-score | 0.8782 | 0.9230 | 0.8995 | 0.8981 | 0.9280 | 0.8271 | 0.4616 | 0.8694 | **0.9280** |
| Transaction | AUROC | 0.9085 | 0.9437 | 0.9216 | 0.9202 | 0.9461 | 0.8773 | 0.7162 | 0.9245 | **0.9543** |
| | AUPRC | 0.3722 | 0.4811 | 0.4520 | 0.3961 | 0.5063 | 0.3688 | 0.0927 | 0.4376 | **0.5642** |
| | Recall@k | 0.3410 | 0.5172 | 0.4462 | 0.3730 | 0.5217 | 0.3730 | 0.0915 | 0.4577 | **0.5835** |
| | F1-score | 0.6502 | 0.7469 | 0.7088 | 0.6552 | 0.7762 | 0.6138 | 0.4105 | 0.6785 | **0.7944** |

where $f_c$ is the frequency of class $c$ in training set, and $\epsilon$ is for numerical stability. To adjust the gradients based on Ricci Flow, we further define the Ricci Flow adjustment term for the $i$-th sample:

$$\Delta \boldsymbol{p}_{i,c} = -\gamma \kappa_c \nabla_{\boldsymbol{p}_{i,c}} \mathcal{L}_{CE},$$

where $\gamma$ is a hyperparameter, and the RFACE can be defined as:

$$\mathcal{L}_{RFACE} = -\frac{1}{N} \sum_{i=1}^{N} \sum_{c=1}^{C} \boldsymbol{y}_{i,c} \log(\tilde{\boldsymbol{p}}_{i,c}),$$

where $\tilde{\boldsymbol{p}}_{i,c} = \text{Sigmoid}(\boldsymbol{z}_{i,c} + \Delta \boldsymbol{p}_{i,c})$. The following theorems proves the benefits of utilizing RFACE as the training objective for heterogeneous GAD, demosntrated in Appendix A:

**Theorem 8.** *For a rare class $c$, $|\nabla_{\boldsymbol{z}_{i,c}} \mathcal{L}_{RFACE}| > |\nabla_{\boldsymbol{z}_{i,c}} \mathcal{L}_{CE}|$ with amplifying factor proportional to $\gamma |\kappa_c|$ and the amplification follows $(1 + \gamma |\kappa_c|)$-Lipschitz continuous, preserving the topology of the latent graph embedding space.*

**Theorem 9.** *When the adjusted predictions are perfect, i.e., $\tilde{\boldsymbol{p}}_{i,c} = \boldsymbol{y}_{i,c}, \forall i, c$, the adjustment term vanishes, i.e., $\Delta \boldsymbol{p}_{i,c} \to 0$, and the raw predictions also converge to the true labels, i.e., $\boldsymbol{p}_{i,c} \to \boldsymbol{y}_{i,c}$ for all classes, including rare ones.*

In summary, the RFACE is a dynamic system that actively recalibrates the learning focus based on per-class performance, not a simple weighting scheme. This ensures our JPGNN is optimized for detecting rare anomalies, making the entire pipeline, from feature projection and topology fusion to spectral filtering, coherent and optimal for the task. Beyond its theoretical foundation, extensive experiments in Section 5 confirm the practical superiority of JacobiGAD.

## 5 EXPERIMENTS

### 5.1 EXPERIMENTAL SETUP

**Datasets and baselines.** We evaluate JacobiGAD on 14 public and 1 private real-world datasets, divided into 20%/20%/60% for train/validation/test, and compare our JacobiGAD with 18 baselines

Table 2: Average performance with multiple runs (heterogeneous graph classification models).

| Datasets | Metrics | HMGNN | muxGNN | HeGCL | RFAGNN | SHGLNN | JacobiGAD |
|---|---|---|---|---|---|---|---|
| SF-295 | AUROC | 0.4112 | 0.4348 | 0.6584 | 0.6799 | 0.5088 | **0.7729** |
| | AUPRC | 0.0421 | 0.0417 | 0.1129 | 0.1090 | 0.0471 | **0.2623** |
| | Recall@k | 0.0477 | 0.0230 | 0.1745 | 0.1802 | 0.0198 | **0.3210** |
| | F1-score | 0.4871 | 0.4871 | 0.4876 | 0.4871 | 0.4871 | **0.6356** |
| SN12C | AUROC | 0.4309 | 0.5169 | 0.6373 | 0.6958 | 0.5053 | **0.7797** |
| | AUPRC | 0.0410 | 0.0784 | 0.0894 | 0.1077 | 0.0458 | **0.2666** |
| | Recall@k | 0.0315 | 0.1347 | 0.1287 | 0.1577 | 0.0188 | **0.3240** |
| | F1-score | 0.4892 | 0.5383 | 0.4874 | 0.4920 | 0.4875 | **0.6329** |
| UACC257 | AUROC | 0.5512 | 0.4207 | 0.6835 | 0.7129 | 0.5022 | **0.7613** |
| | AUPRC | 0.0698 | 0.0332 | 0.1305 | 0.1297 | 0.0381 | **0.1995** |
| | Recall@k | 0.1014 | 0.0142 | 0.2049 | 0.1755 | 0.0132 | **0.2819** |
| | F1-score | 0.4895 | 0.4895 | 0.4962 | 0.4953 | 0.4895 | **0.6246** |
| DBLP | AUROC | 0.4849 | 0.9697 | 0.9696 | 0.9814 | 0.7684 | **0.9830** |
| | AUPRC | 0.3786 | 0.9697 | 0.9698 | 0.9826 | 0.5949 | **0.9842** |
| | Recall@k | 0.3602 | 0.9172 | 0.9306 | 0.9530 | 0.5996 | **0.9575** |
| | F1-score | 0.5079 | 0.9388 | 0.9522 | 0.9595 | 0.6763 | **0.9651** |
| IMDB | AUROC | 0.5256 | 0.6176 | 0.6512 | 0.6594 | 0.5220 | **0.7263** |
| | AUPRC | 0.6063 | 0.6716 | 0.7033 | 0.7119 | 0.5978 | **0.7619** |
| | Recall@k | 0.5983 | 0.6572 | 0.6709 | 0.6909 | 0.5889 | **0.7192** |
| | F1-score | 0.3682 | 0.5171 | 0.6089 | 0.6214 | 0.3682 | **0.6585** |
| PDNS | AUROC | 0.5563 | 0.6250 | 0.7796 | 0.7359 | 0.5173 | **0.8728** |
| | AUPRC | 0.2115 | 0.2528 | 0.4190 | 0.3977 | 0.1912 | **0.6871** |
| | Recall@k | 0.2420 | 0.2855 | 0.4583 | 0.3980 | 0.2470 | **0.6283** |
| | F1-score | 0.4525 | 0.5289 | 0.5432 | 0.6194 | 0.4526 | **0.7760** |
| RCDD | AUROC | 0.7105 | 0.9523 | 0.9390 | 0.9809 | 0.5819 | **0.9826** |
| | AUPRC | 0.2848 | 0.8470 | 0.8031 | 0.9219 | 0.1531 | **0.9332** |
| | Recall@k | 0.2911 | 0.7870 | 0.7366 | 0.8645 | 0.0739 | **0.8747** |
| | F1-score | 0.4625 | 0.8610 | 0.8404 | 0.9182 | 0.4614 | **0.9280** |
| Transaction | AUROC | 0.6409 | 0.8415 | 0.8853 | 0.9338 | 0.5745 | **0.9543** |
| | AUPRC | 0.0773 | 0.3302 | 0.3781 | 0.4331 | 0.0533 | **0.5642** |
| | Recall@k | 0.0572 | 0.3021 | 0.3753 | 0.4348 | 0.0984 | **0.5835** |
| | F1-score | 0.4118 | 0.6241 | 0.6447 | 0.6690 | 0.3504 | **0.7944** |

Table 3: Average performance with multiple runs (GAD models).

| Datasets | Metrics | iGAD | GmapAD | RumorMixer | RQGNN | UniGAD | JacobiGAD |
|---|---|---|---|---|---|---|---|
| SF-295 | AUROC | 0.6768 | 0.6190 | 0.4092 | 0.7657 | 0.5947 | **0.7729** |
| | AUPRC | 0.1040 | 0.0670 | 0.0414 | 0.1938 | 0.0724 | **0.2623** |
| | Recall@k | 0.1531 | 0.0724 | 0.0280 | 0.2683 | 0.1095 | **0.3210** |
| | F1-score | 0.5427 | 0.4095 | 0.4871 | 0.6154 | 0.4971 | **0.6356** |
| SN12C | AUROC | 0.7416 | 0.5957 | 0.3549 | 0.7695 | 0.6281 | **0.7797** |
| | AUPRC | 0.1581 | 0.0605 | 0.0353 | 0.1973 | 0.0769 | **0.2666** |
| | Recall@k | 0.2242 | 0.0733 | 0.0290 | 0.2558 | 0.1151 | **0.3240** |
| | F1-score | 0.5476 | 0.3477 | 0.4875 | 0.5844 | 0.4756 | **0.6329** |
| UACC257 | AUROC | 0.7404 | 0.5936 | 0.4997 | 0.7599 | 0.5973 | **0.7613** |
| | AUPRC | 0.1323 | 0.0507 | 0.0411 | 0.1894 | 0.0672 | **0.1995** |
| | Recall@k | 0.2140 | 0.0527 | 0.0456 | 0.2465 | 0.1176 | **0.2819** |
| | F1-score | 0.5429 | 0.3461 | 0.4895 | 0.6064 | 0.5058 | **0.6246** |
| DBLP | AUROC | 0.9791 | 0.5551 | 0.5000 | 0.9804 | 0.9644 | **0.9830** |
| | AUPRC | 0.9803 | 0.4131 | 0.3835 | 0.9829 | 0.9541 | **0.9842** |
| | Recall@k | 0.9396 | 0.4452 | 0.3792 | 0.9463 | 0.8881 | **0.9575** |
| | F1-score | 0.9598 | 0.5479 | 0.2772 | 0.9509 | 0.9122 | **0.9651** |
| IMDB | AUROC | 0.6530 | 0.5079 | 0.4989 | 0.6707 | 0.6528 | **0.7263** |
| | AUPRC | 0.6971 | 0.5866 | 0.5822 | 0.7254 | 0.6902 | **0.7619** |
| | Recall@k | 0.6909 | 0.5794 | 0.5783 | 0.6845 | 0.6982 | **0.7192** |
| | F1-score | 0.6313 | 0.5073 | 0.3681 | 0.6294 | 0.6332 | **0.6585** |
| PDNS | AUROC | 0.8502 | 0.5173 | 0.5928 | 0.7550 | 0.7310 | **0.8728** |
| | AUPRC | 0.6399 | 0.1810 | 0.2606 | 0.4109 | 0.4293 | **0.6871** |
| | Recall@k | 0.5870 | 0.1999 | 0.2732 | 0.4309 | 0.4146 | **0.6283** |
| | F1-score | 0.7308 | 0.5058 | 0.4526 | 0.5420 | 0.6240 | **0.7760** |
| RCDD | AUROC | 0.9794 | 0.7895 | 0.7335 | 0.9624 | 0.9561 | **0.9826** |
| | AUPRC | 0.9225 | 0.3065 | 0.3134 | 0.8740 | 0.8636 | **0.9332** |
| | Recall@k | 0.8611 | 0.3288 | 0.4513 | 0.8073 | 0.7960 | **0.8747** |
| | F1-score | 0.9190 | 0.6463 | 0.4614 | 0.8874 | 0.8541 | **0.9280** |
| Transaction | AUROC | 0.9431 | 0.7384 | 0.6422 | 0.9353 | 0.8862 | **0.9543** |
| | AUPRC | 0.4626 | 0.0873 | 0.0992 | 0.4939 | 0.3505 | **0.5642** |
| | Recall@k | 0.4897 | 0.0709 | 0.1030 | 0.4966 | 0.3501 | **0.5835** |
| | F1-score | 0.7152 | 0.3875 | 0.4213 | 0.7422 | 0.6307 | **0.7944** |

in the related area. Details can be found in Appendix B. Due to the limited space, we present results of 7 public and 1 private datasets in Section 5, and those of the other 7 public datasets in Appendix E.

**Experimental Settings.** We ensure a fair evaluation by standardizing our approach: baseline models use code from GitHub and their authors' recommended hyperparameters. Note that, since the most commonly used three GNNs, GCN, SAGE, and GAT, are designed for node classification tasks, we

thus implement them with Pytorch_Geometric package and the weighted Cross-Entropy Loss, using the default hyperparameters. JacobiGAD's hyperparameters are rigorously tuned via grid search to maximize validation performance (summed AUROC/AUPRC/Recall@k/F1-score). Configurations are listed in Appendix D.

## 5.2 EXPERIMENTAL RESULTS

We conduct a comprehensive comparison of JacobiGAD against three major groups of competing methods: 8 widely used homogeneous graph classification models, 5 representative heterogeneous graph classification approaches, and 5 novel graph-level anomaly detection methods. The results across 8 datasets are summarized in Tables 1, 2, and 3. We elaborate on our findings in detail next.

To begin with, Table 1 demonstrates that JacobiGAD consistently surpasses classical homogeneous GNN architectures, including GCN, SAGE, GAT, and GIN. These baselines, although foundational, remain surprisingly competitive compared with several more advanced techniques. Remarkably, newer homogeneous GNNs, such as LRGNN, GRDL, UQGNN, and UIL, do not perform as well, frequently falling behind even the simpler models. Their limited performance on heterogeneous graph anomaly detection can be attributed to two main issues: they cannot adaptively integrate information across multiple semantic views well, and they lack mechanisms to properly address the severe class imbalance inherent in GAD tasks.

We then compare JacobiGAD with contemporary heterogeneous graph classification methods, including HMGNN, muxGNN, HeGCL, RFAGNN, and SHGLNN. As shown in Table 2, JacobiGAD consistently yields better detection accuracy across all datasets. Although these models are designed specifically for heterogeneous graphs, their representation learning pipelines often rely on fixed or suboptimal strategies for combining heterogeneous modalities, limiting their expressiveness. Thus, Such drawbacks may distort the graph information, especially when running on complex real-world heterogeneous graphs, leading to sometimes inferior performance, even compared to state-of-the-art homogeneous models. Moreover, most of them do not explicitly mitigate data imbalance, which is especially detrimental in anomaly detection scenarios where abnormal samples are extremely scarce.

Finally, we benchmark against the dedicated graph anomaly detection methods iGAD, GmapAD, RumorMixer, RQGNN, and UniGAD. Their comparative performance, reported in Table 3, indicates that JacobiGAD achieves substantially stronger detection capability. These GAD models are tailored for specific anomaly settings, primarily in homogeneous graphs, and therefore struggle with our target task. Their architectures generally lack the capacity to jointly capture multi-view semantic signals and the high-order structural irregularities that characterize anomalies in heterogeneous graphs. Consequently, even though they are specialized for anomaly detection, their design inherently limits their applicability in the heterogeneous graph setting considered in this work.

## 5.3 ABLATION STUDY

We further examine the influence of key components in JacboGAD, i.e., $\mathcal{L}_{RFACE}$, tunable Jacobi Polynomial parameters $\alpha, \beta$, and learnable view weights $\omega_r$. As shown in Table 4, the ablation study demonstrates the critical contribution of each proposed component to the overall performance of the JacobiGAD. Using $\mathcal{L}_{CE}$ to replace $\mathcal{L}_{RFACE}$ results in significant and consistent performance degradation across all datasets, underscoring its vital role in effectively tackling imbalanced issues in heterogeneous GAD tasks. The learnable parameters $(\alpha, \beta)$ of Jacobi Polynomials also prove essential, as the fix of them leads to a clear decline in performance, which shows that, without a flexible enough graph filter, the model can not handle the complex information within heterogeneous graphs. Similarly, the learnable relation weight $\omega_r$ contributes positively, with its fixing causing noticeable dips, demonstrating the importance of adaptive weights for different relations in the heterogeneous graphs. To sum up, the full model consistently outperforms all ablated variants, confirming that all three components work in concert to achieve state-of-the-art anomaly detection performance across diverse heterogeneous graph datasets.

## 5.4 HYPERPARAMETER ANALYSIS

Figure 2 reports the AUROC, AUPRC, Recall@k, and F1-score of JacobiGAD on the RCDD dataset as we vary $\eta, h_{\dim}, K, T, \epsilon, \gamma$, where $\eta$ is the learning rate, $h_{\dim}$ is the hidden dimension of

Table 4: Ablation study for component deactivation.

| Datasets | Metrics | JacobiGAD | w/o $\mathcal{L}_{RFACE}$ | w/o learnable $(\alpha, \beta)$ | w/o learnable $\omega_r$ |
|---|---|---|---|---|---|
| SF-295 | AUROC | 0.7729 | 0.7591 | 0.7727 | 0.7574 |
| | AUPRC | 0.2623 | 0.1953 | 0.2172 | 0.1940 |
| | Recall@k | 0.3210 | 0.2815 | 0.2905 | 0.2667 |
| | F1-score | 0.6356 | 0.5984 | 0.6161 | 0.5931 |
| SN12C | AUROC | 0.7797 | 0.7463 | 0.7505 | 0.7651 |
| | AUPRC | 0.2666 | 0.2345 | 0.2119 | 0.2240 |
| | Recall@k | 0.3240 | 0.3035 | 0.2933 | 0.2856 |
| | F1-score | 0.6329 | 0.6129 | 0.5733 | 0.5942 |
| UACC257 | AUROC | 0.7613 | 0.7282 | 0.7365 | 0.7499 |
| | AUPRC | 0.1995 | 0.1617 | 0.1713 | 0.1709 |
| | Recall@k | 0.2819 | 0.2241 | 0.2475 | 0.2535 |
| | F1-score | 0.6246 | 0.5634 | 0.5829 | 0.5647 |
| DBLP | AUROC | 0.9830 | 0.9820 | 0.9745 | 0.9694 |
| | AUPRC | 0.9842 | 0.9831 | 0.9762 | 0.9683 |
| | Recall@k | 0.9575 | 0.9530 | 0.9441 | 0.9418 |
| | F1-score | 0.9651 | 0.9641 | 0.9576 | 0.9605 |
| IMDB | AUROC | 0.7263 | 0.6962 | 0.7116 | 0.7025 |
| | AUPRC | 0.7619 | 0.7313 | 0.7514 | 0.7486 |
| | Recall@k | 0.7192 | 0.7045 | 0.7108 | 0.7014 |
| | F1-score | 0.6585 | 0.6439 | 0.6325 | 0.6414 |
| PDNS | AUROC | 0.8728 | 0.8691 | 0.8728 | 0.8689 |
| | AUPRC | 0.6871 | 0.6749 | 0.6812 | 0.6718 |
| | Recall@k | 0.6283 | 0.6149 | 0.6239 | 0.6195 |
| | F1-score | 0.7760 | 0.7695 | 0.7685 | 0.7583 |
| RCDD | AUROC | 0.9826 | 0.9808 | 0.9814 | 0.9814 |
| | AUPRC | 0.9332 | 0.9325 | 0.9318 | 0.9293 |
| | Recall@k | 0.8747 | 0.8749 | 0.8739 | 0.8675 |
| | F1-score | 0.9280 | 0.9274 | 0.9264 | 0.9237 |

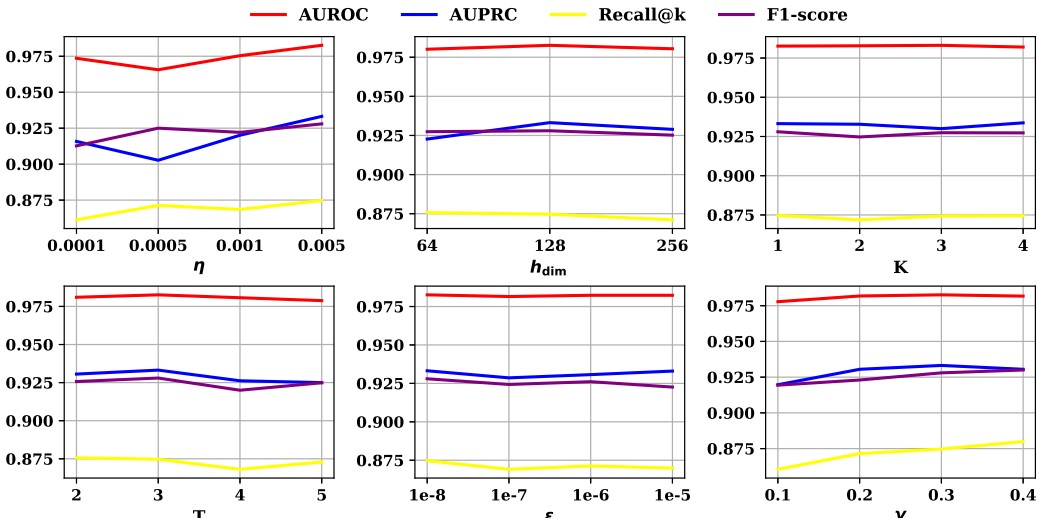

Figure 2: The change of performance on RCDD when varying different hyperparameters.

JacobiGAD, $K, T$ are the width and depth of JacobiGAD respectively, $\epsilon$ is the small value to keep $\kappa_c$ in RFACE valid, and $\gamma$ is the adjusted hyperparameter in RFACE. As shown in Figure 2, JacobiGAD remains stable when varying the hyperparameters, demonstrating its stability.

## 6 CONCLUSION

This paper proposed JacobiGAD, a novel framework for heterogeneous GAD. Our approach integrates a theoretically grounded random projection for feature alignment, a Jacobi Polynomial-based spectral GNN for superior multi-view fusion and cross-geometric representation learning, and a Ricci Flow-inspired loss that dynamically counteracts class imbalance. Supported by strong theoretical guarantees and extensive experimental validation, JacobiGAD establishes a new state-of-the-art, providing a powerful and principled methodology for GAD on complex heterogeneous graphs.

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

## A  PROOF

**Proof of Theorem 1.** Suppose we have $N$ vectors $\boldsymbol{v}_1, \boldsymbol{v}_2, \cdots, \boldsymbol{v}_N$, where $\boldsymbol{v}_i \in \mathbb{R}^{d_i}$. We first prove the equivalence of different zero-padding methods, namely end padding, front padding, and scatter padding. Let $D = \max_i d_i$, then we have:

- End padding: $g(\boldsymbol{v}_i) = (\boldsymbol{v}_{i1}, \boldsymbol{v}_{i2}, \cdots, \boldsymbol{v}_{id_i}, 0, 0, c \ldots, 0)$,
- Front padding: $g(\boldsymbol{v}_i) = (0, 0, c \ldots, 0, \boldsymbol{v}_{i1}, \boldsymbol{v}_{i2}, \cdots, \boldsymbol{v}_{id_i})$,
- Scatter padding: Randomly generate a subset $I_i$ of $\{1, 2, \cdots, D\}$, where $|I_i| = d_i$, then

$$(g(\boldsymbol{v}_i))_j = \begin{cases} \boldsymbol{v}_k, j = \text{the } k\text{-th element of } I_i, \\ 0, j \notin I_i \end{cases}$$

All of these $g(\cdot)$ are linear isometries, i.e., $||g(\boldsymbol{u}) - g(\boldsymbol{v})|| = ||\boldsymbol{u} - \boldsymbol{v}||$, which means the zero padding ways are equivalent for preserving distance. □

Denote $\boldsymbol{x}_i = g(\boldsymbol{v}_i) \in \mathbb{R}^D$ and draw $\boldsymbol{P} \in \mathbb{R}^{d \times D}$ with $\boldsymbol{P}_{ij} \sim \mathcal{N}(0, \frac{1}{d})$, then we have:

$$f : \mathbb{R}^D \to \mathbb{R}^d, f(\boldsymbol{x}) = \boldsymbol{P}\boldsymbol{x}$$

Consider the random variable

$$\boldsymbol{X} = ||f(\boldsymbol{u})||_2^2 = \sum_{k=1}^{d} \langle \boldsymbol{P}_{k,*}, \boldsymbol{u} \rangle^2,$$

where $\boldsymbol{u} = \boldsymbol{x}_i - \boldsymbol{x}_j$, and $\boldsymbol{P}_{k,*}$ is the $k$-th row of $\boldsymbol{P}$, we have each inner product $\langle \boldsymbol{P}_{k,*}, \boldsymbol{u} \rangle$ is Gaussian with mean 0 and variance

$$\text{Var}(\langle \boldsymbol{P}_{k,*}, \boldsymbol{u} \rangle) = \sum_{l=1}^{D} \text{Var}(\boldsymbol{P}_{k,l}) \boldsymbol{u}_l^2 = \frac{1}{d} \sum_{l=1}^{D} \boldsymbol{u}_l^2 = \frac{||\boldsymbol{u}||_2^2}{d}$$

Hence we have:

$$\boldsymbol{Y}_k = \sqrt{\frac{d}{||\boldsymbol{u}||_2^2}} \langle \boldsymbol{P}_{k,*}, \boldsymbol{u} \rangle \sim \mathcal{N}(0, 1),$$

$$\boldsymbol{X} = \sum_{k=1}^{d} \langle \boldsymbol{P}_{k,*}, \boldsymbol{u} \rangle^2 = \frac{||\boldsymbol{u}||_2^2}{d} \sum_{k=1}^{d} \boldsymbol{Y}_k^2$$

Therefore, $\boldsymbol{Z} = \sum_{k=1}^{d} \boldsymbol{Y}_k^2$ is $\chi^2$ with $d$ degrees of freedom, and $\boldsymbol{X} = \frac{||\boldsymbol{u}||_2^2}{d} \boldsymbol{Z}$.

A standard inequality (Zhang & Zhou, 2020) for tail bound of $\chi^2$ random variable demonstrates:

$$\Pr[|\boldsymbol{X} - \boldsymbol{u}||_2^2| \geq \epsilon \boldsymbol{u}||_2^2] \leq 2e^{-\frac{\epsilon^2 d}{4}}, \forall 0 < \epsilon < 1$$

Then for any $\boldsymbol{u}$, we can have:

$$\Pr[(1 - \epsilon)\boldsymbol{u}||_2^2 \leq ||\boldsymbol{P}\boldsymbol{u} \leq (1 + \epsilon)\boldsymbol{u}||_2^2||] \geq 1 - 2e^{-\frac{\epsilon^2 d}{4}}$$

We care about $M$ pairs of vectors, then by the union bound, the probability that all $M$ pairs of distances are preserved is at least $1 - 2Me^{-\frac{\epsilon^2 d}{4}}$. □

**Proof of Theorem 2.** The optimization process of a spectral GNN in graph classification can be defined as:

$$\text{argmin}_{\theta_k, \boldsymbol{W}} \mathcal{L}(y, \text{Pooling}(\sum_{k=1}^{K} \theta_k g_k(\boldsymbol{L}) \boldsymbol{X} \boldsymbol{W}))$$

For simplicity, let $\mathcal{L}$ be the MSE loss function, and the pooling function be the mean pooling function. Then we can reformulate the process for $i$-th sample with $n_i$ nodes as:

$$\text{argmin}_{\theta_k, \boldsymbol{W}} \frac{1}{2} (\boldsymbol{p}^T \sum_{k=1}^{K} \theta_k g_k(\boldsymbol{L}) \boldsymbol{X} \boldsymbol{W} - y)^2,$$

where $\boldsymbol{p}^T \in \mathbb{R}^{1 \times n_i}$ represents the mean pooling function vector with each entry as $\frac{1}{n_i}$. Then, over a dataset of $N$ graphs, we define the targeted process after reordering as:

$$\text{argmin}_{\theta_k, \boldsymbol{W}} \frac{1}{2N} \sum_{i=1}^{N} (\sum_{k=1}^{K} \theta_k a_k^{(i)} - y^{(i)})^2,$$

where $a_k^{(i)} = (\boldsymbol{p}^{(i)})^T g_k(\boldsymbol{L}^{(i)}) \boldsymbol{X}^{(i)} \boldsymbol{W}$.

According to previous work (Wang & Zhang, 2022), the learned filter function is nearly identical across different bases since they share the same expressive power and can all reach the global minimum. Therefore, the optimization of $\boldsymbol{W}$ is largely independent of the basis selection near the global minimum. In contrast, the optimization of $\theta_k$ is significantly influenced by the choice of basis. To emphasize the impact of basis selection, we will focus exclusively on the optimization of $\theta_k$.

To analyze the convergence speed near the global minimum, we then derive the Hessian matrix $\boldsymbol{H}$ of the process with respect to $\theta_k$:

$$\boldsymbol{H}_{jk} = \frac{1}{N} \sum_{i=1}^{N} a_j^{(i)} a_k^{(i)}$$

Diagonalize each Laplacian matrix $\boldsymbol{L}^{(i)} = \boldsymbol{U}^{(i)} \boldsymbol{\Lambda}^{(i)} (\boldsymbol{U}^{(i)})^T$, we can get:

$$a_k^{(i)} = \sum_{l=1}^{n_i} g_k(\lambda_l^{(i)}) \phi_l^{(i)} \psi_l^{(i)},$$

where $\lambda_l^{(i)}$ is the $l$-th eigenvalue of $\boldsymbol{L}^{(i)}$, $\phi_l^{(i)} = [(\boldsymbol{U}^{(i)})^T \boldsymbol{X}^{(i)} \boldsymbol{W}]_l$, and $\psi_l^{(i)} = \boldsymbol{p}^{(i)} \boldsymbol{u}_l^{(i)}$.

Since the Laplacian matrix is the normalized Laplacian matrix, the eigenvalue distribution of all graphs converges to a density $\rho(\lambda)$ on $[0, 2]$. Assume the random coefficients $\phi_l^{(i)}, \psi_l^{(i)}$ decorrelate between different $l$, and $\mathbb{E}[(\phi\psi)^2 | \lambda]$ depends only on $\lambda$. We have:

$$\frac{1}{N} \sum_{i=1}^{N} a_j^{(i)} a_k^{(i)} \xrightarrow{N \to \infty} \int_0^2 g_j(\lambda) g_k(\lambda) \omega(\lambda) d\lambda,$$

where $\omega(\lambda) = \rho(\lambda) \mathbb{E}[(\phi\psi)^2 | \lambda]$. In other words, we have:

$$\boldsymbol{H} \propto [\langle g_j, g_k \rangle_\omega]_{j,k=0}^{K},$$

where $\langle g_j, g_k \rangle_\omega = \int_0^2 g_j(\lambda) g_k(\lambda) \omega(\lambda) d\lambda$. Reaching the global minimum means $\boldsymbol{H}$ is a diagonal matrix ($\langle g_j, g_k \rangle_\omega = 0$, where $j \neq k$), which is equivalent to that $g(\cdot)$ is an orthonormal basis in the polynomial space. Therefore, we choose a general form of orthogonal polynomials with flexible enough weight functions to adapt to different graph signal density functions, i.e., Jacobi Polynomials. □

**Proof of Theorem 3.** We first prove the injectivity of the combination of the Jacobi Polynomial-based graph filter to show that it can preserve the full information from the original graph.

Suppose we have a combination of Jacobi Polynomial-based graph filter, that is, $\mathcal{T}(\boldsymbol{x}) = \sum_{v=1}^{V} P_n^{(\alpha_v, \beta_v)}(\boldsymbol{L}_v) \boldsymbol{x}$, where $\boldsymbol{x}$ is a graph signal, $P_n^{(\alpha_v, \beta_v)}$ is the Jacobi Polynomial-based graph filter for the $v$-th view, and $\boldsymbol{L}_v$ is the Laplacian matrix for $v$-th view. Define the kernel characterization for $\mathcal{T}$:

$$\text{ker}(\mathcal{T}) = \{\boldsymbol{x} \in \mathbb{R}^N | \sum_{v=1}^{V} P_n^{(\alpha_v, \beta_v)}(\boldsymbol{L}_v) \boldsymbol{x} = 0\}$$

To prove the injectivity is equal to prove $\text{ker}(\mathcal{T}) = \{0\}$ under $V$ heterogeneous views.

Assume the different views of a heterogeneous graph satisfy the following condition:

- Spectral Disjointness: $\forall i \neq j, \text{Eigen}(\boldsymbol{L}_i) \cap \text{Eigen}(\boldsymbol{L}_j) =$, i.e., there are no shared eigenvalues between Laplacian matrices of different views.
- Full Spectral Coverage: $\cup_{v=1}^{V} \text{Eigen}(\boldsymbol{L}_v) = \mathbb{R}_{\geq 0}$, i.e., eigenvalues of Laplacian matrix cover the entire spectrum.

- No Common Eigenvectors: $\not\exists \boldsymbol{x} \neq 0, \boldsymbol{L}_v \boldsymbol{x} = \lambda_v \boldsymbol{x}, \forall v$, i.e., different Laplacian matrices have distinct eigenspaces.

By tuning $\alpha_v, \beta_v$ of each $P_n^{(\alpha_v, \beta_v)}$, we can easily obtain root avoidance of the Jacobi Polynomial-based graph filter:

$$P_n^{(\alpha_v, \beta_v)}(\lambda) \neq 0, \forall \lambda \in \text{Eigen}(\boldsymbol{L}_v),$$

as $P_n^{(\alpha_v, \beta_v)}(\boldsymbol{x})$ have $n$ real roots in $[-1, 1]$, and the spectra $\boldsymbol{L}_v$ can be rescale to $[-1, 1]$.

By root avoidance and spectral disjointness, we have:

$$P_n^{(\alpha_v, \beta_v)}(\lambda) \neq 0 \Rightarrow |P_n^{(\alpha_v, \beta_v)}(\lambda)| > \delta_v > 0,$$

where $\delta_v$ is a small value larger than 0, which shows the strict positivity of $|P_n^{(\alpha_v, \beta_v)}(\lambda)|$.

Then for $\boldsymbol{x} \neq 0$, we expand the eigenbasis of each $\boldsymbol{L}_v$:

$$\boldsymbol{x} = \sum_{i=1}^{N} c_{v,i} \boldsymbol{u}_{v,i},$$

where $c_{v,i} = \boldsymbol{u}_{v,i}^T \boldsymbol{x}$, $\boldsymbol{u}_{v,i}$ is the $i$-th eigenvector of $\boldsymbol{L}_v$, and for $a \neq b$, $\boldsymbol{u}_{a,c}^T \boldsymbol{u}_{b,d} = 0, \forall b, d$.

Therefore, we can define the $v$-th filtered component as:

$$\boldsymbol{z}_v = P_n^{(\alpha_v, \beta_v)}(\boldsymbol{L}_v)\boldsymbol{x} = \sum_{i=1}^{N} P_n^{(\alpha_v, \beta_v)}(\lambda_{v,i}) c_{v,i} \boldsymbol{u}_{v,i}$$

Thus, we can compute:

$$||\mathcal{T}(\boldsymbol{x})||_2^2 = \langle \sum_{i=0}^{V} \boldsymbol{z}_i, \sum_{j=0}^{V} \boldsymbol{z}_j \rangle = \sum_{v=1}^{V} ||\boldsymbol{z}_v||_2^2 + \sum_{i \neq j} \langle \boldsymbol{z}_i, \boldsymbol{z}_j \rangle,$$

where $\langle \boldsymbol{z}_i, \boldsymbol{z}_j \rangle = \boldsymbol{z}_i^T \boldsymbol{z}_j = 0, \forall i \neq j$, due to orthogonality.

By spectral coverage, $\exists v$ and $i$ such that:

$$|c_{v,i}| > 0, \lambda_{v,i} \in \text{Eigen}(\boldsymbol{L}_v),$$

for which view $v$ and $i$ we have:

$$||\boldsymbol{z}_v||_2^2 \geq |P_n^{(\alpha_v, \beta_v)}(\lambda_{v,i})|^2 |c_{v,i}|^2 > \delta_v^2 |c_{v,i}|^2 > 0$$

Thus, we have:

$$||\mathcal{T}(\boldsymbol{x})||_2^2 > 0 \Rightarrow \mathcal{T}(\boldsymbol{x}) \neq 0,$$

which means strict positivity. Then we have $\ker(\mathcal{T}) = \{0\}$, i.e., injectivity, as desired. $\square$

**Proof of Theorem 4.** We define the fused feature extractor on the target component as:

$$\mathcal{F}(\boldsymbol{x}) = \sum_{v=1}^{V} \Pi_{\mathcal{S}} P_n^{(\alpha_v^*, \beta_v^*)}(\boldsymbol{L}_v)\boldsymbol{x},$$

where $\Pi_{\mathcal{S}}$ is the projection onto the target feature subspace, and $P_n^{(\alpha_v^*, \beta_v^*)}$ is the optimized Jacobi Polynomial filter for view $v$. Then the enhancement factor is:

$$\gamma(\mathcal{F}) = \min_{\boldsymbol{z} \in \mathcal{S}, ||\boldsymbol{z}||_2^2 = 1} ||\mathcal{F}(\boldsymbol{z})||_2^2,$$

which represents the worst-case amplification of target features by the fused extractor.

Assume that:

- The target subspace $\mathcal{S}$ is spannded by common eigenvectors of all graph Laplacians $\boldsymbol{L}_v$, which means for each $\boldsymbol{z} \in \mathcal{S}$, $\exists i$ such that $\boldsymbol{L}_v \boldsymbol{z} = \lambda_{v,i} \boldsymbol{z}, \forall v$, where $\lambda_{v,i}$ is the $i$-th eigenvalue of $\boldsymbol{L}_v$.

- For each view $v$, the Jacobi Polynomial filter is designed such that for all eigenvalues $\lambda_{v,i}$ associated with $\mathcal{S}$, we have $\forall \boldsymbol{z} \in \mathcal{S}, P_n^{(\alpha_v^*, \beta_v^*)}(\boldsymbol{L}_v)\boldsymbol{z} = c_v\boldsymbol{z}$, where $c_v = P_n^{(\alpha_v^*, \beta_v^*)}(\lambda_{v,i}) > 0$.
- There exists a constant $c_{\min} > 0$ such that $c_v \geq c_{\min}, \forall v$.

Thus, for any $\boldsymbol{z} \in \mathcal{S}$ with $||\boldsymbol{z}||_2^2 = 1$, we have:

$$\Pi_{\mathcal{S}} P_n^{(\alpha_v^*, \beta_v^*)}(\boldsymbol{L}_v)\boldsymbol{z} = \Pi_{\mathcal{S}}(c_v\boldsymbol{z}) = c_v\boldsymbol{z},$$

since $\Pi_{\mathcal{S}}\boldsymbol{z} = \boldsymbol{z}(\boldsymbol{z} \in \mathcal{S})$.

Then we can have:

$$\mathcal{F}(\boldsymbol{z}) = \sum_{v=1}^{V} c_v\boldsymbol{z} = (\sum_{v=1}^{V} c_v)\boldsymbol{z},$$

whose norm is:

$$||\mathcal{F}(\boldsymbol{z})||_2^2 = ||(\sum_{v=1}^{V} c_v)\boldsymbol{z}||_2^2 = |(\sum_{v=1}^{V} c_v)|||\boldsymbol{z}||_2^2 = \sum_{v=1}^{V} c_v$$

Give that $c_v \geq c_{\min} > 0, \forall v$, we have:

$$\gamma(\mathcal{F}) = \min_{\boldsymbol{z} \in \mathcal{S}, ||\boldsymbol{z}||_2^2 = 1} ||\mathcal{F}(\boldsymbol{z})||_2^2 \geq Vc_{\min},$$

which means $\gamma(\mathcal{F}) = \Theta(V)$. $\qquad\square$

Next, we show its robustness to noise. Consider a signal $\boldsymbol{x}$ with target component $\boldsymbol{x}_{target} \in \mathcal{S}$ and noise $\boldsymbol{e} \sim \mathcal{N}(0, \sigma^2\boldsymbol{I})$ uncorrelated across views. Then we have:

$$\gamma(\mathcal{F}) = \mathcal{F}(\boldsymbol{x}_{target}) + \mathcal{F}(\boldsymbol{e})$$

From the above proof, we have the following for the target term:

$$||\mathcal{F}(\boldsymbol{x}_{target})||_2^2 \geq \gamma(\mathcal{F})||\boldsymbol{x}_{target}||_2^2 \geq Vc_{\min}||\boldsymbol{x}_{target}||_2^2$$

For the noise term, we have:

$$\mathbb{E}[||\mathcal{F}(\boldsymbol{e})||_2^2] = \mathbb{E}[||\sum_{v=1}^{V} \Pi_{\mathcal{S}} P_n^{(\alpha_v^*, \beta_v^*)}(\boldsymbol{L}_v)\boldsymbol{e}||_2^2]$$

$$= \sum_{v=1}^{V} \mathbb{E}[||\Pi_{\mathcal{S}} P_n^{(\alpha_v^*, \beta_v^*)}(\boldsymbol{L}_v)\boldsymbol{e}||_2^2]$$

$$\leq \sum_{v=1}^{V} \sigma^2 ||\Pi_{\mathcal{S}} P_n^{(\alpha_v^*, \beta_v^*)}(\boldsymbol{L}_v)||_F^2,$$

since the noise is uncorrelated across views.

Assume $||\Pi_{\mathcal{S}} P_n^{(\alpha_v^*, \beta_v^*)}(\boldsymbol{L}_v)||_F^2 \leq M$ for some constant $M$, we have:

$$\mathbb{E}[||\mathcal{F}(\boldsymbol{e})||_2^2] \leq V\sigma^2 M$$

By Jensen's inequality, we have:

$$\mathbb{E}[||\mathcal{F}(\boldsymbol{e})||_2^2] \leq \sqrt{\mathbb{E}[||\mathcal{F}(\boldsymbol{e})||_2^2]} \leq \sigma\sqrt{VM}$$

Thus, the signal-to-noise ratio is:

$$\frac{||\mathcal{F}(\boldsymbol{x}_{target})||_2^2}{\mathbb{E}[||\mathcal{F}(\boldsymbol{e})||_2^2]} \geq \frac{Vc_{\min}||\boldsymbol{x}_{target}||_2^2}{\sigma\sqrt{VM}} = \frac{\sqrt{V}c_{\min}||\boldsymbol{x}_{target}||_2^2}{\sigma\sqrt{M}} = \Theta(\sqrt{V})$$

$\qquad\square$

**Proof of Theorem 5.** The $\kappa$-stereographic model provides a unified framework for Euclidean, Hyperbolic, and Spherical geometries through a common metric, parameterized by the curvature $\kappa$:

$$ds^2 = \frac{dr^2 + r^2 d\Omega^2}{(1 + \kappa r^2)^2},$$

where $s$ is the square of an infinitesimally small distance between two points in a space, $r$ is the radial coordinate in stereographic projection, and $d\Omega^2$ is the metric on the unit sphere.

Jacobi Polynomials $P_n^{(\alpha,\beta)}(x)$ are orthogonal polynomials on $[-1,1]$ with respect to the weight $(1-x)^\alpha(1+x)^\beta$. They arise as eigenfunctions of the Laplace-Beltrami operator in the $\kappa$-stereographic model for specific values of $\kappa$ and after appropriate coordinate transformations. Below, we derive the connections for each geometry.

For $\kappa > 0$, the space is sperical with raidus $R = \frac{1}{\sqrt{\kappa}}$. The Laplace-Beltrami operator $\Delta$ in stereographic coordinates has eigenfunctions that can be separated into radial and angular parts. The Laplace-Beltrami operator for a radial function $f(r)$ in $n$ dimensions is:

$$\Delta f = \kappa \frac{(1+\kappa r^2)^n}{r^{n-1}} \frac{d}{dr}[r^{n-1}(1+\kappa r^2)^{2-n}\frac{df}{dr}],$$

whose eigenvalue equation is $\Delta f + \lambda f = 0$.

To solve the equation, we introduce $u = \kappa r^2 \geq 0$:

$$\frac{df}{dr} = 2\sqrt{\kappa u}\frac{df}{du}, \frac{d}{dr} = 2\sqrt{\kappa u}\frac{d}{du}$$

Then the equation can be represented as:

$$\frac{(1+u)^n}{u^{\frac{n-1}{2}}} \frac{d}{du}[u^{\frac{n}{2}}(1+u)^{2-n}\frac{df}{du}] + \frac{\lambda}{4\kappa^2}f = 0$$

Change variable to $x = \frac{u-1}{u+1} = \frac{\kappa r^2 - 1}{\kappa r^2 + 1}$, so $x \in [-1,1]$. Then $u = \frac{1+x}{1-x}$ and the derivatives become:

$$\frac{d}{du} = \frac{dx}{du}\frac{d}{dx} = -\frac{2}{(1-x)^2}\frac{d}{dx}, \frac{d^2}{du^2} = \frac{4}{(1-x)^4}\frac{d^2}{dx^2} + \frac{8}{(1-x)^3}\frac{d}{dx}$$

Then the equation can be simplifies to the Jacobi differential equation:

$$(1-x^2)\frac{d^2f}{dx^2} + [\beta - \alpha - (\alpha + \beta + 2)x]\frac{df}{dx} + k(k + \alpha + \beta + 1)f = 0,$$

where $\alpha = \beta = \frac{n-2}{2}$, and $k$ is the quantum number related to the eigenvalue $\lambda = k(k + n - 1)\kappa$.

The solutions are Jacobi Polynomials:

$$f(r) \propto P_k^{(\frac{n-2}{2},\frac{n-2}{2})}(\frac{\kappa r^2 - 1}{\kappa r^2 + 1})$$

$\square$

For $\kappa < 0$, the sapce is hyperbolic with curvature radius $R = \frac{1}{\sqrt{|\kappa|}}$. The spectrum of the Laplace-Beltrami operator is continuous, and the radial eigenfunctions are not polynomials but can be expressed as Jacobi functions (analytical continuations of Jacobi Polynomials).

Set $\kappa = -|\kappa|$, so $1 + \kappa r^2 = 1 - |\kappa|r^2$. The eigenvalue equation is similar to the spherical case:

$$\Delta f + \lambda f = 0, \Delta f = \kappa \frac{(1+\kappa r^2)^n}{r^{n-1}} \frac{d}{dr}[r^{n-1}(1+\kappa r^2)^{2-n}\frac{df}{dr}]$$

Then we use $u = |\kappa|r^2$ to get:

$$\frac{(1+u)^n}{u^{\frac{n-1}{2}}} \frac{d}{du}[u^{\frac{n}{2}}(1+u)^{2-n}\frac{df}{du}] + \frac{\lambda}{4\kappa^2}f = 0$$

Change variable to $x = \frac{u-1}{u+1} = \frac{|\kappa|r^2 - 1}{|\kappa|r^2 + 1}$, so $x \in [-\infty, 0]$, then the equation becomes a confluent hypergeometric equation, which is solved by Jacobi functions with parameters $\alpha = \beta = \frac{n-2}{2}$:

$$f(r) \propto P_{-\frac{1}{2}+i\sigma}^{(\frac{n-2}{2},\frac{n-2}{2})}(\frac{|\kappa|r^2 - 1}{|\kappa|r^2 + 1}),$$

where $\sigma = \sqrt{\frac{\lambda}{|\kappa|} - \frac{(n-1)^2}{4}}, \forall \lambda > \frac{(n-1)^2 |\kappa|}{4}$, and $i$ is the imaginary unit. □

For $\kappa = 0$, the space is Euclidean space. The Laplace-Beltrami operator reduces to the standard Laplacian, and the radial eigenfunctions are Bessel functions, which arise as a limit of Jacobi Polynomials as $\kappa \to 0$.

When $\kappa \to 0$, the metric is $ds^2 = dr^2 + r^2 d\Omega^2$, and the radial eigenvalue equation is:

$$\frac{1}{r^{n-1}} \frac{d}{dr}(r^{n-1} \frac{df}{dr}) + \lambda f = 0$$

This is the spherical Bessel equation, whose solutions are Bessel functions:

$$f(r) \propto r^{-\frac{n-2}{2}} J,$$

where $J$ stands for $J_n(x) = \sum_{m=0}^{\infty} \frac{(-1)^m}{m! \Gamma(m+n+1)}(\frac{x}{2})^{2m+n}$, and $\Gamma(\cdot)$ is the gamma function.

For fixed $r$ and $k$, as $\kappa \to 0$, the Jacobi Polynomial limit is:

$$\lim_{\kappa \to 0} P_k^{(\frac{n-2}{2}, \frac{n-2}{2})}(\frac{\kappa r^2 - 1}{\kappa r^2 + 1}) \propto r^{-\frac{n-2}{2}} J_{\frac{n-2}{2}}(\sqrt{k(k+n-1)}r),$$

□

**Proof of Theorem 6.** First, we introduce the following theorem:

**Theorem 10** (Weierstrass approximation theorem). *Suppose $f(x)$ is a continuous real-valued function defined on the real interval $[a, b]$. For every $\epsilon > 0$, there exist a polynomial $p$ such that for all $x$ in $[a, b]$, we have $||f(x) - p(x)||_\infty < \epsilon$.*

Then, given a $f(x) \in C([-1, 1])$ and $\epsilon > 0$, by theorem 10, we can pick a genuine polynomial $p(x) = \sum_{i=1}^{N} w_i x^i$ so that $||f(x) - p(x)||_\infty < \epsilon$.

And by definition, each Jacobi Polynomial $P_n^{(\alpha, \beta)}(x)$ is a genuine polynomial of degree $n$, so:

$$\text{Span}(P_0^{(\alpha,\beta)}, P_1^{(\alpha,\beta)}, \cdots) = \{\text{all real polynomials in } x\},$$

which means every polynonmial $p(x)$ of degree $\leq N$ can be written uniquely in the Jacobi basis:

$$p(x) = \sum_{i=1}^{N} c_i P_i^{(\alpha,\beta)}(x)$$

Hence the finite Jacobi Polynomial sum $S_N(x) = \sum_{i=1}^{N} c_i P_i^{(\alpha,\beta)}(x)$ satisfies:

$$||f(x) - S_N(x)||_\infty = ||f(x) - p(x)||_\infty < \epsilon$$

Thus, any continuous $f$ on $[-1, 1]$ can be uniformly approximated by finite linear combinations of Jacobi polynomials. □

Then we consider the weighted space $L_w^2[-1, 1]$, where the weight function is the same as Jacobi Polynomials:

$$w(x) = (1 - x)^\alpha (1 + x)^\beta,$$

with $\alpha, \beta > -1$.

Let $\mu$ be the measure defined by $d\mu = w(x)dx$. Since $\alpha, \beta > -1$, the integral satisfies:

$$\int_{-1}^{1} w(x)dx < \infty,$$

which means $\mu$ is a finite Borel measure on $[-1, 1]$.

Denote any function $q \in L_w^2[-1, 1]$, we then prove that Jacobi Polynomials can approximate it in the $L_w^2$ norm.

Since $\mu$ is a finite Borel measure on the compact interval $[-1, 1]$, the continuous functions on $[-1, 1]$ area dense in $L_w^2[-1, 1]$, which is proved in (). Thus, for any $\delta > 0$, there exists a continuous function $t$ on $[-1, 1]$ such that:

$$||q(x) - t(x)||_{L_w^2} < \frac{\delta}{2}$$

Then, following the previous proof, we have:

$$||t(x) - p(x)|| = \sup_{x \in [-1,1]} |t(x) - p(x)| < \epsilon,$$

where $p(x)$ is finite linear combinations of Jacobi polynomials.

For any $\epsilon > 0$, we can choose $\epsilon$, such that:

$$\epsilon^2 \int_{-1}^{1} w(x)dx < (\frac{\delta}{2})^2$$

Thus, we have:

$$||t(x) - p(x)||_{L_w^2}^2 = \int_{-1}^{1} |t(x) - p(x)|^2 w(x)dx \le \epsilon^2 \int_{-1}^{1} w(x)dx < (\frac{\delta}{2})^2,$$

which means $||t(x) - p(x)||_{L_w^2} < \frac{\delta}{2}$.

By triangle inequality, we can derive:

$$||q(x) - p(x)||_{L_w^2} \le ||q(x) - t(x)||_{L_w^2} + ||t(x) - p(x)||_{L_w^2} < \frac{\delta}{2} + \frac{\delta}{2} = \delta,$$

which concludes that Jacobi Polynomials can approximate any function $q \in L_w^2[-1,1]$. $\qquad\square$

**Proof of Theorem 7.** Let $E_N(f(x)) = \min_{\deg(g(x)) \le N} ||f(x) - g(x)||_\infty$, where $f(x) \in C^{r-1}([-1,1])$, $f^{(r)}(x) \in L(w^{(\alpha,\beta)}(x))$, $r$ is the derivative order, and $w^{(\alpha,\beta)}(x) = (1-x)^\alpha (1+x)^\beta$.

We then construct a positive kernel:

$$K_N^{(r)}(x,t) = \sum_{k=0}^{N} a_{N,k} P_k^{(\alpha,\beta)}(x) P_k^{(\alpha,\beta)}(t),$$

where $a_{N,k} \ge 0$ chosen so that for each fixed $x \in [-1,1]$:

- Normalization: $\int_{-1}^{1} K_N^{(r)}(x,t) w^{(\alpha,\beta)}(t)dt = 1$,
- Moment vanishing up to order $r-1$: $\int_{-1}^{1} (t-x)^m K_N^{(r)}(x,t) w^{(\alpha,\beta)}(t)dt = 0, m = 1, 2, \cdots, r-1$,
- High-order moment bound: $\int_{-1}^{1} |t-x|^r K_N^{(r)}(x,t) w^{(\alpha,\beta)}(t)dt \le \frac{C}{N^r}$, where $C$ depends on $r, \alpha, \beta$

The existence of such a kernel is standard, i.e., the classical Jackson kernel in orthogonal-polynomial theory (Rudin, 1987).

Then define the Jackson operator $J_N$ by:

$$(J_N f)(x) = \int_{-1}^{1} f(t) K_N^{(r)}(x,t) w^{(\alpha,\beta)}(t)dt,$$

where $J_N f(x)$ is a polynomial of degree $\le N$, because $K_N^{(r)}(x,\cdot)$ is a sum of Jacobi Polynomials up to degree $N$. Consequently:

$$E_N(f(x)) \le ||f(x) - (J_N f)(x)||_\infty$$

Next, we use repeated integration of the Taylor formula. For each $t \in [-1,1]$, there holds:

$$f(t) = f(x) + f'(x)(t-x) + \cdots + \frac{f^{(r-1)}(x)}{(r-1)!}(t-x)^{r-1} + R_r(x,t),$$

where the remainder can be written in integral form:

$$R_r(x,t) = \frac{1}{(r-1)!} \int_0^1 (1-u)^{r-1} f^{(r)}(x + u(t-x))(t-x)^r du$$

By the moment-vanishing property of $K_N^{(r)}(x,t)$, when we subtract out the Taylor part, all terms up to $(t-x)^{r-1}$ integrate to 0. Thus:

$$f(x) - (J_N f)(x) = \int_{-1}^{1} [f(t) - \text{Taylor at } x] K_N^{(r)}(x,t) w^{(\alpha,\beta)}(t) dt$$

$$= \int_{-1}^{1} R_r(x,t) K_N^{(r)}(x,t) w^{(\alpha,\beta)}(t) dt$$

Insert the integral form of $R_r(x,t)$:

$$f(x) - (J_N f)(x) = \frac{1}{(r-1)!} \int_{-1}^{1} \int_{0}^{1} (1-u)^{r-1} f^{(r)}(x+u(t-x))(t-x)^r K_N^{(r)}(x,t) w^{(\alpha,\beta)}(t) du dt$$

Taking absolute values and using Fubini's theorem:

$$|f(x) - (J_N f)(x)| \le \frac{1}{(r-1)!} \int_{-1}^{1} (1-u)^{r-1} \int_{0}^{1} |f^{(r)}(x+u(t-x))| |t-x|^r K_N^{(r)}(x,t) w^{(\alpha,\beta)}(t) dt du$$

Then, change variables in the inner integral, i.e., for each fixed $u$, the map $t \to s = x + u(t-x)$ is linear of Jacobian $dt = \frac{ds}{u}$. Moreover $w^{(\alpha,\beta)}(t) dt \le C w^{(\alpha,\beta)}(s) ds$, since $w$ is smooth and $u \in [0,1]$. One shows:

$$|t-x|^r K_N^{(r)}(x,t) w^{(\alpha,\beta)}(t) dt \le \frac{C}{N^r} w^{(\alpha,\beta)}(s) ds,$$

by the high-order moment bound. Hence, we have:

$$|f(x) - (J_N f)(x)| \le \frac{C}{N^r} \int_{0}^{1} (1-u)^{r-1} \frac{du}{u} \int_{-1}^{1} |f^{(r)}(s)| w^{(\alpha,\beta)}(s) ds,$$

where $\int_{0}^{1} (1-u)^{r-1} \frac{du}{u}$ converges to a constant depending only on $r$. We conclude:

$$||f(x) - (J_N f)(x)||_\infty \le |f(x) - (J_N f)(x)| \le \frac{C}{N^r} \int_{-1}^{1} |f^{(r)}(s)| w^{(\alpha,\beta)}(s) ds,$$

which completes the proof:

$$E_N(f(x)) \le ||f(x) - (J_N f)(x)||_\infty \le \frac{C}{N^r} \int_{-1}^{1} |f^{(r)}(t)| w^{(\alpha,\beta)}(t) dt = \frac{C}{N} ||f^{(r)}(x)||_{L_w^1},$$

where $J_N f$ is a polynomial of degree $\le N$ and $C$ depends only on $r, \alpha, \beta$. $\qquad\square$

**Proof of Theorem 8.** For simplicity, we assume the total class number $C$ is 2.

We first calculate the gradient flow of $\mathcal{L}_{RFACE}$ using the chain rule:

$$\nabla_{\boldsymbol{z}_{i,c}} \mathcal{L}_{RFACE} = (\tilde{\boldsymbol{p}}_{i,c} - \boldsymbol{y}_{i,c})[1 - \gamma \kappa_c \boldsymbol{p}_{i,c}(1 - \boldsymbol{p}_{i,c})]$$

Since $\kappa_c < 0$:

$$\gamma|\kappa_c|\boldsymbol{p}_{i,c}(1 - \boldsymbol{p}_{i,c}) > 0 \Rightarrow [1 + \gamma|\kappa_c|\boldsymbol{p}_{i,c}(1 - \boldsymbol{p}_{i,c})] > 1$$

We then calculate the gradient flow of $\mathcal{L}_{CE}$ using the chain rule:

$$\nabla_{\boldsymbol{z}_{i,c}} \mathcal{L}_{CE} = \boldsymbol{p}_{i,c} - 1$$

We focus on the case when $\boldsymbol{y}_{i,c} = 1, \boldsymbol{p}_{i,c} \approx 0$ (probability of anomalous class is low):

$$|\tilde{\boldsymbol{p}}_{i,c} - \boldsymbol{y}_{i,c}| \approx |\boldsymbol{p}_{i,c} - 1|$$

Then we can have:

$$|\nabla_{\boldsymbol{z}_{i,c}} \mathcal{L}_{RFACE}| = [1 + \gamma|\kappa_c|\boldsymbol{p}_{i,c}(1 - \boldsymbol{p}_{i,c})]|\boldsymbol{p}_{i,c} - 1| > |\nabla_{\boldsymbol{z}_{i,c}} \mathcal{L}_{CE}| = |\boldsymbol{p}_{i,c} - 1|,$$

with an amplifying factor $[1 + \gamma|\kappa_c|\boldsymbol{p}_{i,c}(1 - \boldsymbol{p}_{i,c})]$ proportional to $\gamma|\kappa_c|$. $\qquad\square$

Next, we prove the adjustment $\Delta\boldsymbol{p}_{i,c}$ is $(1 + \gamma|\kappa_c|)$-Lipschitz continuous, so the topology of the latent graph embedding space is preserved.

Define $g(z_{i,c}) = \Delta \boldsymbol{p}_{i,c} = -\gamma \kappa_c (\boldsymbol{p}_{i,c} - \boldsymbol{y}_{i,c})$, then for two different graphs embedding $\boldsymbol{z}_1, \boldsymbol{z}_2$, we have:

$$|g(\boldsymbol{z}_1) - g(\boldsymbol{z}_2)| = |-\gamma \kappa_c (\boldsymbol{p}_1 - \boldsymbol{p}_2)| = \gamma |\kappa_c| |\boldsymbol{p}_1 - \boldsymbol{p}_2|$$

Since Sigmoid is 1-Lipschitz, we have:

$$|g(\boldsymbol{z}_1) - g(\boldsymbol{z}_2)| \leq \gamma |\kappa_c| |\boldsymbol{z}_1 - \boldsymbol{z}_2|,$$

which means $g(\cdot)$ is $\gamma |\kappa_c|$-Lipschitz.

Then the adjusted logit $\tilde{\boldsymbol{z}}_1 = \boldsymbol{z}_1 + g(\boldsymbol{z}_1), \tilde{\boldsymbol{z}}_2 = \boldsymbol{z}_2 + g(\boldsymbol{z}_2)$ satisfies:

$$|\tilde{\boldsymbol{z}}_1 - \tilde{\boldsymbol{z}}_2| \leq |\boldsymbol{z}_1 - \boldsymbol{z}_2| + |g(\boldsymbol{z}_1) - g(\boldsymbol{z}_2)| \leq (1 + \gamma |\kappa_c|) |\boldsymbol{z}_1 - \boldsymbol{z}_2|,$$

which means the amplification follows $(1 + \gamma |\kappa_c|)$-Lipschitz. $\qquad \square$

**Proof of Theorem 9.** When $\tilde{\boldsymbol{p}}_{i,c} = \boldsymbol{y}_{i,c}$, there are two cases, i.e., $\boldsymbol{y}_{i,c} = 0$ and $\boldsymbol{y}_{i,c} = 1$.

For $\boldsymbol{y}_{i,c} = 0$, we have:

$$\tilde{\boldsymbol{p}}_{i,c} = 0 \Rightarrow \text{Sigmoid}(\boldsymbol{z}_{i,c} + \Delta \boldsymbol{p}_{i,c}) = 0 \Rightarrow \boldsymbol{z}_{i,c} + \Delta \boldsymbol{p}_{i,c} \to -\infty$$

In this situation, since $\boldsymbol{z}_{i,c} + \Delta \boldsymbol{p}_{i,c} = \boldsymbol{z}_{i,c} - \gamma \kappa_c \boldsymbol{p}_{i,c}$, and $-\gamma \kappa_c \boldsymbol{p}_{i,c}$ is bounded, $\boldsymbol{z}_{i,c} + \Delta \boldsymbol{p}_{i,c} \to -\infty$ requires $\boldsymbol{z}_{i,c} \to -\infty$, which implies $\boldsymbol{p}_{i,c} = \text{Sigmoid}(\boldsymbol{z}_{i,c}) \to 0 = \boldsymbol{y}_{i,c}$.

For $\boldsymbol{y}_{i,c} = 1$, we have:

$$\tilde{\boldsymbol{p}}_{i,c} = 1 \Rightarrow \text{Sigmoid}(\boldsymbol{z}_{i,c} + \Delta \boldsymbol{p}_{i,c}) = 1 \Rightarrow \boldsymbol{z}_{i,c} + \Delta \boldsymbol{p}_{i,c} \to \infty$$

In this situation, since $\boldsymbol{z}_{i,c} + \Delta \boldsymbol{p}_{i,c} = \boldsymbol{z}_{i,c} - \gamma \kappa_c (\boldsymbol{p}_{i,c} - 1)$, and $-\gamma \kappa_c (\boldsymbol{p}_{i,c} - 1)$ is bounded, $\boldsymbol{z}_{i,c} + \Delta \boldsymbol{p}_{i,c} \to \infty$ requires $\boldsymbol{z}_{i,c} \to \infty$, which implies $\boldsymbol{p}_{i,c} = \text{Sigmoid}(\boldsymbol{z}_{i,c}) \to 1 = \boldsymbol{y}_{i,c}$.

$\qquad \square$

## B   DATASETS AND BASELINES

**Datasets.** MCF-7, MOLT-4, PC-3, SW-620, NCI-H23, OVCAR-8, P388, SF-295, SN12C, and UACC257 are 10 small-molecule biological activity datasets from TUDataset (Morris et al., 2020), each corresponding to a different cancer cell line screen. Compounds are represented as heterogeneous graphs where nodes are atom types and edges are the bonds between them. Remarkably, we utilize the original data in the TUDataset datasets, where the number of node and edge types are large, as the real chemical compounds are extremely complex. The large number of node and edge types in the public datasets posts additional challenges for the heterogeneous graph-level anomaly detection. Each compound is labeled as active or inactive against its respective cancer type; we treat inactive compounds as normal and active ones as anomalies. Node features are one-hot encodings of the atom labels.

The above public datasets are originally graph classification datasets, whereas the datasets below are node classification datasets. Therefore, we need to transform them into graph classification datasets. The transformation is the same:

- We follow the original anomalous ratio to sample $n_n$ normal nodes and $n_a$ anomalous nodes from a heterogeneous graph, where $\frac{n_a}{n_n + n_a}$ is the anomalous ratio of the original dataset, to simulate the imbalanced nature of graph-level anomaly detection tasks.
- Use original Breadth-First Search algorithm to obtain the subgraph around $n_n$ normal nodes and $n_a$ anomalous nodes. For small-scale graphs, such as DBLP and IMDB, we set small $n_n$ and $n_a$ and 3 as the sampling layer number to limit the overlap between graphs, while keep enough information in each graph. For large-scale graphs, such as PDNS and RCDD, we set large $n_n$ and $n_a$ and 2 as the sampling layer number to provide diverse enough samples while reduce the running cost for several baselines for fair comparison, because some of them might cost high computational resources, as reported in Appendix C.
- Use the center node label as the subgraph label.

DBLP and IMDB are public datasets processed by pytorch_geometric (Fey & Lenssen, 2019). The DBLP dataset is a subset of the computer science bibliography, comprising four node types: authors,

Table 5: Datasets used in the experiments, where $n_n, n_a$ are the normal and anomalous number of graphs respectively, $r = \frac{n_a}{n_n+n_a}$ is the anomalous rate of the dataset, $n, m$ are the average number of nodes and edges in graphs respectively, $T_V, R_E$ are the types of nodes and edges in graphs respectively, and $d$ is the dimension of nodes after projection.

| Source | Type | Dataset | $n_n$ | $n_a$ | $r$ | $n$ | $m$ | $T_V$ | $R_E$ | $d$ |
|---|---|---|---|---|---|---|---|---|---|---|
| Public | Bioinfo | MCF-7 | 25476 | 2294 | 0.0826 | 26.40 | 28.53 | 46 | 129 | 46 |
| | | MOLT-4 | 36625 | 3140 | 0.0790 | 26.10 | 28.14 | 64 | 176 | 64 |
| | | PC-3 | 25941 | 1568 | 0.0570 | 26.36 | 28.49 | 45 | 133 | 45 |
| | | SW-620 | 38122 | 2410 | 0.0595 | 26.06 | 28.09 | 65 | 184 | 65 |
| | | NCI-H23 | 38296 | 2057 | 0.0510 | 26.07 | 28.10 | 65 | 182 | 65 |
| | | OVCAR-8 | 38437 | 2079 | 0.0513 | 26.08 | 28.11 | 65 | 184 | 65 |
| | | P388 | 39174 | 2298 | 0.0554 | 22.11 | 23.56 | 72 | 271 | 72 |
| | | SF-295 | 38246 | 2025 | 0.0503 | 26.06 | 28.09 | 65 | 184 | 65 |
| | | SN12C | 38049 | 1955 | 0.0489 | 26.08 | 28.11 | 65 | 184 | 65 |
| | | UACC257 | 38345 | 1643 | 0.0411 | 26.09 | 28.13 | 64 | 176 | 64 |
| | Citation | DBLP | 1197 | 745 | 0.3836 | 162.08 | 96.29 | 4 | 6 | 50 |
| | Social | IMDB | 1584 | 1135 | 0.4174 | 85.35 | 59.52 | 3 | 4 | 64 |
| | Cybersecurity | PDNS | 41337 | 8663 | 0.1733 | 48.80 | 72.13 | 2 | 3 | 32 |
| | Finance | RCDD | 50000 | 8364 | 0.1433 | 17.43 | 9.20 | 7 | 8 | 256 |
| Private | | Transaction | 20000 | 437 | 0.0214 | 14.87 | 20.12 | 6 | 11 | 325 |

papers, terms, and conferences. Authors are categorized by research area (database, data mining, artificial intelligence, information retrieval) and are represented by a bag-of-words feature vector derived from their paper keywords. For our task, authors from the database area are designated as normal nodes, while those from data mining are treated as anomalous. The IMDB dataset is a subset of the Internet Movie Database, containing movies, actors, and directors as node types. Movies are classified by genre (action, comedy, drama) and are represented by bag-of-words features from their plot keywords. In this context, action movies are the normal class, and comedy movies are the anomalous class.

PDNS and RCDD are public datasets collected from Kaggle [1]. The PDNS dataset is a cybersecurity graph constructed from a seed set of malicious domains. Its infrastructure data is extracted from a global passive DNS repository. The graph contains two entity types (domains and IPs) connected by four relations (e.g., "domain resolves to IP"). Each domain node has a 10-dimensional feature vector derived from its domain name and a binary label identifying it as malicious. We directly use these original labels to define normal and anomalous nodes. The RCDD is a large-scale e-commerce network from Alibaba, built for real-world risk detection. It contains 7 node types (e.g., buyer, seller) and 7 edge types (e.g., buy, sell), though specific names are anonymized for confidentiality. In this network, risk nodes often disguise themselves by forging relationships. Each node is described by a 256-dimensional feature vector, and item nodes are labeled as either risk commodities or normal. These original labels are used to designate the normal and anomalous classes.

The final dataset is a proprietary financial heterogeneous graph provided by a prominent company. Its objective is to identify sub-networks, or communities, associated with suspicious or non-compliant activity. The graph schema is complex, comprising 6 node types (e.g., representing real users and entities) and 11 edge types that define the intricate relationships between them. The task is naturally a GAD problem: each entire graph is labeled as either containing a risky community or being normal. We directly adopt these original labels to train our model to distinguish between anomalous and normal graphs.

**Baselines.** The first group is homogeneous graph classification models:

- **GCN** (Kipf & Welling, 2017): A foundational graph convolutional network that performs neighborhood aggregation through a spectral graph convolution-inspired operation.

- **SAGE** (Hamilton et al., 2017): A scalable inductive framework that generates node embeddings by sampling and aggregating features from a node's local neighborhood.

---

[1]https://www.kaggle.com/

- **GAT** (Velickovic et al., 2018): Employs an attention mechanism to compute hidden representations by assigning different weights to each neighbor node.
- **GIN** (Xu et al., 2019): A theoretically powerful model designed to be as expressive as the Weisfeiler-Lehman graph isomorphism test.
- **LRGNN** (Wei et al., 2023): Addresses the limitation of shallow receptive fields by stacking multiple GNNs to capture long-range dependencies between distant nodes.
- **GRDL** (Wang & Fan, 2024): Treats node embeddings as discrete distributions within a latent space, enabling graph-level classification without a global readout function.
- **UQGNN** (Wu et al., 2025): A model that integrates uncertainty quantification into the graph representation learning process, producing confidence estimates alongside predictions.
- **UIL** (Sui et al., 2025): Provides a unified framework for invariant graph learning by enforcing both structural and semantic invariance, leading to the identification of more robust and stable node representations.

The second group is heterogeneous graph classification models:

- **HMGNN** (Yu & Gao, 2022): Models complex heterogeneous structures by constructing heterogeneous motif graphs to capture rich semantic information from multiple node and edge types.
- **muxGNN** (Melton & Krishnan, 2023): Represents graphs as multiplex networks, using separate graphs for each relation type and a coupling graph to connect node representations across these relations.
- **HeGCL** (Shi et al., 2024): A contrastive learning framework that learns node and graph embeddings by contrasting a meta-path view with a global network topology view.
- **RFAGNN** (Wu et al., 2024): Handles both heterophily and heterogeneity within a unified model using a relation-based frequency adaptive graph filter.
- **SHGLNN** (Hayat et al., 2024): Leverages hypergraphs constructed from heterogeneous graphs to model complex higher-order (intra- and inter-graph) contextual relationships.

The third group is graph-level anomaly detection models:

- **iGAD** (Zhang et al., 2022): Anomaly detection is performed by comparing input graphs against a set of prototypical neural substructure patterns.
- **GmapAD** (Ma et al., 2023): Maps entire graphs into a well-structured latent space where normal and anomalous graphs are more easily separable.
- **RumorMixer** (Xu et al., 2024): A specialized model for rumor detection that captures the echo chamber effect and platform heterogeneity inherent in social networks.
- **RQGNN** (Dong et al., 2024): Leverages the Rayleigh Quotient to combine spectral and spatial information for anomaly detection.
- **UniGAD** (Lin et al., 2024): A unified framework that integrates node-level, subgraph-level, and graph-level information for comprehensive graph anomaly detection.

## C  ALGORITHM AND COMPLEXITY

We first analyze the Preprocess function. As shown in Algorithm 1, in lines 1-6, we have in total of $O(N)$, where $N$ is the number of graphs in the dataset, as we need to find the $d_{\max}$ of all the graphs. Then, in lines 7-11, we need to do the projection for each graph in the dataset. Each will cost $O(Vdd_{max})$, where $V$ is the number of nodes in graph $G$. Hence, the total cost will be $O(Nndd_{\max})$, where $n$ is the average number of nodes in each graph of $\mathcal{G}$. Therefore, the total time complexity of Preprocess is $O(Nndd_{\max})$.

Then, we analyze the time complexity of JPGNN for each graph $G$. As presented in Algorithm 2, in lines 1-8, the dominant cost is the summation of weighted adjacency matrices. In practice, we don't need the summation, as we can multiply the coefficients by the edge weights. Thus, the total cost is $O(E)$, where $E$ is the number of edges in the graph. Then, for lines 9-16, the dominant cost should be line 12, which has a cost of $O(KTVEd_{hid}^3)$, where $d_{hid}$ is the hidden dimension of the layer of the GNN. Therefore, the total time complexity of JPGNN is $O(KTVEd_{hid}^3)$. Next, we analyze the time complexity of RFACE in Algorithm 3. In lines 1-6, we only need to use basic operations with $O(1)$ time complexity. Therefore, the total time complexity of RFACE is $O(1)$.

**Algorithm 1:** Preprocess

**Input:** $\mathcal{G}, d$
**Output:** $\mathcal{G}'$

1  $\mathcal{G}' \leftarrow \mathcal{G}$;
2  $d_{\max} \leftarrow 0$;
3  **for** $G$ *in* $\mathcal{G}$ **do**
4     **for** $\boldsymbol{X}_t$ *in* $G.\mathcal{X}$ **do**
5         $d_{\max} \leftarrow \max(d_{\max}, d_t)$;

6  $\boldsymbol{P} \leftarrow \boldsymbol{P}_{i,j} \sim \mathcal{N}(0, \frac{1}{d}), \boldsymbol{P} \in \mathbb{R}^{d \times d_{\max}}$;
7  **for** $G$ *in* $\mathcal{G}'$ **do**
8     $G.\boldsymbol{X}^{proj} \leftarrow$ Null;
9     **for** $\boldsymbol{X}_t$ *in* $G.\mathcal{X}$ **do**
10         $\boldsymbol{X}_t^{proj} \leftarrow \boldsymbol{X}_t \oplus \boldsymbol{0}, \boldsymbol{0} \in \mathbb{R}^{|V_t| \times (d_{\max} - d_t)}$;
11         $G.\boldsymbol{X}^{proj} \leftarrow G.\boldsymbol{X}^{proj} \oplus \boldsymbol{P}\boldsymbol{X}^{proj}$;

12  Return $\mathcal{G}'$;

---

**Algorithm 2:** JPGNN

**Input:** $\mathcal{A}, \boldsymbol{X}^{proj}, K, T$
**Output:** $\boldsymbol{z}$

1  $\boldsymbol{A} \leftarrow 0$;
2  **for** $\boldsymbol{A}_r$ *in* $\mathcal{A}$ **do**
3     $\boldsymbol{A} \leftarrow \boldsymbol{A} + \omega_r \boldsymbol{A}_r$;
4  $\boldsymbol{L} \leftarrow \boldsymbol{I} - \boldsymbol{D}^{-\frac{1}{2}} \boldsymbol{A} \boldsymbol{D}^{-\frac{1}{2}}$;
5  $\lambda_{\max} \leftarrow \max(\text{Eigen}(\boldsymbol{L}))$;
6  $\hat{\boldsymbol{L}} \leftarrow \frac{2}{\lambda_{\max}} \boldsymbol{L} - \boldsymbol{I}$;
7  $\boldsymbol{H}^{(0)} \leftarrow \sigma(\boldsymbol{X}^{proj} \boldsymbol{W}^{(0)})$;
8  $\boldsymbol{H}^{stack} \leftarrow \boldsymbol{H}^{(0)}$;
9  **for** $k$ *in* $\{1, \cdots, K\}$ **do**
10     $\boldsymbol{H}^{(k)} \leftarrow 0$;
11     **for** $t$ *in* $\{0, \cdots, T\}$ **do**
12         $\boldsymbol{H}^{(k)} \leftarrow \boldsymbol{H}^{(k)} + \theta_t^{(k)} P_t^{(\alpha^{(k)}, \beta^{(k)})}(\hat{\boldsymbol{L}}) \boldsymbol{H}^{(k-1)} \boldsymbol{W}^{(k)}$;
13     $\boldsymbol{H}^{(k)} \leftarrow \sigma(\boldsymbol{H}^{(k)})$;
14     $\boldsymbol{H}^{stack} \leftarrow \boldsymbol{H}^{stack} \oplus \boldsymbol{H}^{(k)}$;
15  $\boldsymbol{H} \leftarrow \sigma(\boldsymbol{H}^{stack} \boldsymbol{W})$;
16  $\boldsymbol{z} \leftarrow \text{Pooling}(\boldsymbol{H})$;
17  Return $\boldsymbol{z}$;

---

Finally, in Algorithm 4, to clearly show the time complexity of each epoch of the training procedure, we combine the above time complexities. As shown in lines 7-10, we need to call JPGNN and RFACE $|\mathcal{G}|$ times, so the time complexity of each epoch of the training procedure of JacobiGAD is $O(NKTnmd_{hid}^3)$, where $m$ is the average number of edges in each graph of $\mathcal{G}$.

Compared to homogeneous graph-level classification, such as GRDL models. Its time complexity for each sample in each training epoch is $C_1 + O(K(n^2 + mn + m^2))$, as reported in their paper, where $C_1$ is the time complexity of the used GNN, $K$ is the number of classes, and $n, m$ are the number of nodes and edges in each graph. Thus, we can easily conclude $O(\text{JacobiGAD}) \leq O(\text{GRDL})$.

Compared to heterogeneous graph-level classification models, such as HeGCL. Its time complexity for each sample in each training epoch is $Q|\mathcal{N}|^2 + |\mathcal{E}| + |\mathcal{E}^\Phi| + |\Phi||\mathcal{N}|$, as reported in their paper, where $Q$ is the number of heads of attention layer, $|\mathcal{N}|, |\mathcal{E}|$ are the number of nodes and edges in

---

**Algorithm 3:** RFACE

**Input:** $\boldsymbol{\kappa}, \boldsymbol{y}, \boldsymbol{z}, \gamma, C$
**Output:** $\mathcal{L}_{RFACE}$

1   $\boldsymbol{p} \leftarrow \text{Sigmoid}(\boldsymbol{z})$;
2   $\Delta \boldsymbol{p}_c \leftarrow -\gamma \boldsymbol{\kappa}_c (\boldsymbol{p}_c - \boldsymbol{y}_c)$;
3   $\tilde{\boldsymbol{p}} \leftarrow \text{Sigmoid}(\boldsymbol{z} + \Delta \boldsymbol{p})$;
4   $\mathcal{L}_{RFACE} \leftarrow 0$;
5   **for** $c$ *in* $\{1, \cdots, C\}$ **do**
6     $\lfloor \quad \mathcal{L}_{RFACE} \leftarrow \mathcal{L}_{RFACE} + \boldsymbol{y}_c \log(\tilde{\boldsymbol{p}}_c)$;
7   Return $\mathcal{L}_{RFACE}$;

---

**Algorithm 4:** JacobiGAD

**Input:** $\mathcal{G}, \boldsymbol{f}, K, T, \gamma, C, \epsilon, d, E$

1   $\mathcal{G}' \leftarrow \text{Preprocess}(\mathcal{G}, d)$;
2   $\mathcal{L}_{RFACE} \leftarrow 0$;
3   $\boldsymbol{\kappa} \leftarrow \boldsymbol{0}$;
4   **for** $c$ *in* $\{1, \cdots, C\}$ **do**
5     $\lfloor \quad \boldsymbol{\kappa}_c \leftarrow \frac{\boldsymbol{f}_c}{\max(\boldsymbol{f}) + \epsilon}$;
6   **for** *epoch in* $1, \cdots, E$ **do**
7     **for** $G$ *in* $\mathcal{G}'$ **do**
8       $\boldsymbol{z} \leftarrow \text{JPGNN}(G.\mathcal{A}, G.\boldsymbol{X}^{proj}, K, T)$;
9       $\mathcal{L}_{RFACE} \leftarrow \mathcal{L}_{RFACE} + \text{RFACE}(\boldsymbol{\kappa}, G.\boldsymbol{y}, \boldsymbol{z}, \gamma, C)$;
10     $\lfloor \quad \mathcal{L}_{RFACE} \leftarrow -\frac{1}{|\mathcal{G}|} \mathcal{L}_{RFACE}$;

---

each graph, $|\Phi|$ is the number of meta-path, and $|\mathcal{E}^\Phi|$ is the number of meta-path-based edges. Thus, we can easily conclude $O(\text{JacobiGAD}) \leq O(\text{HeGCL})$.

Compared to GAD models, such as RQGNN. Its time complexity for each sample in each training epoch is $O(Kqnmd_{hid}^3)$, where $K, q$ are the width and depth of the GNN, $n, m$ are the number of nodes and edges in each graph, and $d_{hid}$ is the hidden dimension of the layer of the GNN. Thus, we can easily conclude $O(\text{JacobiGAD}) \leq O(\text{RQGNN})$.

To sum up, we compare the theoretical time complexity of JacobiGAD with representative previous works in different categories, and conclude that our JacobiGAD has practical cost for real deployment, as its time complexity is less than or equal to the previous works.

Additionally, we further report the runtime and memory cost of JacobiGAD and compare them with all baselines across 3 datasets. The results in Table 6 show that JacobiGAD achieves competitive computational efficiency while maintaining state-of-the-art detection performance.

Empirically, JacobiGAD's training time is faster than most included baselines, and its total GPU memory usage stays within a comparable range. This indicates that the model scales well with both graph size and dataset difficulty. Notably, JacobiGAD maintains SOTA performance while requiring no additional memory-heavy modules. As a result, JacobiGAD provides a favorable trade-off between efficiency and accuracy: it preserves strong anomaly detection capability without incurring substantial computational cost.

These observations confirm that the proposed method is not only effective but also practical for real-world heterogeneous graph-level anomaly detection scenarios where time and memory resources are often constrained.

## D   EXPERIMENTAL SETTINGS

The hyperparameters used for training JacobiGAD are provided in Table 7. The model was tuned through an extensive grid search over the following values: learning rate $\eta \in$

Table 6: Average wall-clock time (s) and total memory cost (MB).

| Cost | Datasets | MCF-7 | | IMDB | | RCDD | |
|---|---|---|---|---|---|---|---|
| Type | Baselines | Memory | Time | Memory | Time | Memory | Time |
| | GCN | 800.88 | 82.09 | 643.80 | 12.06 | 1165.37 | 214.11 |
| | SAGE | 707.21 | 83.63 | 539.50 | 12.16 | 1065.95 | 219.77 |
| | GAT | 842.66 | 101.17 | 645.05 | 18.88 | 1167.25 | 242.17 |
| Homogeneous Graph Classification | GIN | 693.98 | 484.71 | 495.53 | 65.97 | 1027.38 | 1383.77 |
| | LRGNN | 845.38 | 1013.79 | 823.08 | 357.49 | 1344.03 | 1505.59 |
| | GRDL | 812.52 | 734.36 | 614.29 | 234.39 | 1301.55 | 1937.96 |
| | UQGNN | 1199.74 | 229.02 | 944.09 | 319.57 | 1335.56 | 267.89 |
| | UIL | 1177.79 | 332.46 | 918.78 | 60.83 | 1253.44 | 741.10 |
| | HMGNN | 2098.57 | 342.67 | 913.00 | 121.24 | 2629.50 | 492.56 |
| | muxGNN | 5450.38 | 91.28 | 1325.74 | 27.08 | 12548.35 | 405.09 |
| Heterogeneous Graph Classification | HeGCL | 17812.20 | 1933.91 | 1031.01 | 52.65 | 5156.08 | 762.66 |
| | RFAGNN | 1240.74 | 255.83 | 898.17 | 43.41 | 1743.74 | 523.85 |
| | SHGLNN | 653.80 | 571.93 | 799.83 | 105.06 | 1007.95 | 2123.87 |
| | iGAD | 799.52 | 289.79 | 821.83 | 144.45 | 1247.28 | 801.91 |
| | GmapAD | 1335.84 | 1885.73 | 1077.30 | 85.76 | 2073.54 | 33755.93 |
| Graph-level Anomaly Detection | RumorMixer | 739.81 | 6253.59 | 667.43 | 1401.91 | 1192.67 | 12094.97 |
| | RQGNN | 1177.37 | 2151.12 | 1062.88 | 248.91 | 1682.53 | 8817.05 |
| | UniGAD | 964.59 | 162.33 | 1077.91 | 110.22 | 1420.30 | 2013.64 |
| Ours | JacobiGAD | 1265.71 | 136.22 | 803.02 | 26.67 | 1250.07 | 170.07 |

Table 7: Hyperparameters for different datasets, where $\eta$ is learning rate, $h_{\text{dim}}$ is hidden dimension of JPGNN layers, and $K, T$ are the width and depth of the JPGNN.

| Dataset | $\eta$ | $h_{\text{dim}}$ | $K$ | $T$ |
|---|---|---|---|---|
| MCF-7 | 0.005 | 128 | 3 | 5 |
| MOLT-4 | 0.0001 | 256 | 4 | 5 |
| PC-3 | 0.001 | 128 | 2 | 5 |
| SW-620 | 0.005 | 64 | 4 | 4 |
| NCI-H23 | 0.001 | 256 | 3 | 5 |
| OVCAR-8 | 0.0005 | 256 | 4 | 5 |
| P388 | 0.0001 | 128 | 4 | 4 |
| SF-295 | 0.001 | 256 | 3 | 4 |
| SN12C | 0.001 | 256 | 3 | 4 |
| UACC257 | 0.0005 | 64 | 4 | 4 |
| DBLP | 0.0001 | 256 | 2 | 5 |
| IMDB | 0.005 | 128 | 2 | 5 |
| PDNS | 0.001 | 128 | 3 | 3 |
| RCDD | 0.005 | 128 | 1 | 3 |
| Transaction | 0.001 | 64 | 1 | 4 |

$\{0.005, 0.001, 0.0005, 0.0001\}$, hidden dimension size $h_{\text{dim}} \in \{64, 128, 256\}$, $K \in \{1, 2, 3, 4\}$, and $T \in \{2, 3, 4, 5\}$. The optimal hyperparameter set was chosen based on the best composite performance, considering AUROC, AUPRC, Recall@k, and F1-score, on the validation set, and we report the test results for this configuration. Note that for hyperparameters in RFACE, i.e., $\epsilon$ and $\gamma$, we set them as default values $1e - 8$ and $0.3$ respectively, as they reach a relatively better performance. All trials were executed on an NVIDIA Quadro RTX 8000 to maintain a consistent experimental environment.

# E    ADDITIONAL EXPERIMENTAL RESULTS

To further demonstrate the robustness and generality of our method, we conduct additional experiments on 7 public graph benchmarks: MCF-7, MOLT-4, PC-3, SW-620, NCI-H23, OVCAR-8, and P388. Evaluating on this expanded set enables a more rigorous assessment of our model's ability to generalize across different graph distributions.

Across all datasets, our method consistently outperforms representative homogeneous graph classification baselines, shown in Table 8, heterogeneous graph classification approaches, shown in

Table 8: Average performance with multiple runs (homogeneous graph classification models).

| Datasets | Metrics | GCN | SAGE | GAT | GIN | LRGNN | GRDL | UQGNN | UIL | JacobiGAD |
|---|---|---|---|---|---|---|---|---|---|---|
| MCF-7 | AUROC | 0.6720 | 0.7264 | 0.6971 | 0.7110 | 0.7070 | 0.5349 | 0.5238 | 0.7162 | **0.7679** |
| | AUPRC | 0.1460 | 0.2318 | 0.1983 | 0.1730 | 0.2434 | 0.1115 | 0.0878 | 0.2367 | **0.3403** |
| | Recall@k | 0.1808 | 0.2818 | 0.2585 | 0.2150 | 0.2825 | 0.1590 | 0.0792 | 0.2716 | **0.3769** |
| | F1-score | 0.4783 | 0.5949 | 0.5832 | 0.4872 | 0.5458 | 0.4785 | 0.4785 | 0.5503 | **0.6597** |
| MOLT-4 | AUROC | 0.6628 | 0.7155 | 0.7076 | 0.6880 | 0.7334 | 0.5257 | 0.5856 | 0.6867 | **0.7381** |
| | AUPRC | 0.1367 | 0.2051 | 0.2288 | 0.1611 | 0.2353 | 0.1208 | 0.1016 | 0.1735 | **0.3097** |
| | Recall@k | 0.1624 | 0.2649 | 0.2797 | 0.2033 | 0.2962 | 0.1948 | 0.1200 | 0.2171 | **0.3519** |
| | F1-score | 0.4803 | 0.5903 | 0.6100 | 0.4954 | 0.4906 | 0.4794 | 0.4794 | 0.4794 | **0.6507** |
| PC-3 | AUROC | 0.6717 | 0.7157 | 0.7391 | 0.7109 | 0.7389 | 0.5106 | 0.5395 | 0.7352 | **0.7677** |
| | AUPRC | 0.1017 | 0.1869 | 0.1973 | 0.1321 | 0.2289 | 0.0797 | 0.0693 | 0.2008 | **0.3064** |
| | Recall@k | 0.1318 | 0.2359 | 0.2657 | 0.1679 | 0.3050 | 0.1360 | 0.0903 | 0.2370 | **0.3603** |
| | F1-score | 0.4941 | 0.5882 | 0.6064 | 0.4853 | 0.5120 | 0.4853 | 0.4853 | 0.5229 | **0.6394** |
| SW-620 | AUROC | 0.7000 | 0.7619 | 0.7187 | 0.7270 | 0.7660 | 0.5601 | 0.5525 | 0.7202 | **0.7728** |
| | AUPRC | 0.1342 | 0.2137 | 0.1737 | 0.1371 | 0.2281 | 0.0996 | 0.0654 | 0.1484 | **0.2697** |
| | Recall@k | 0.1715 | 0.2697 | 0.2172 | 0.1853 | 0.2918 | 0.1362 | 0.0539 | 0.1777 | **0.3347** |
| | F1-score | 0.4990 | 0.5880 | 0.5695 | 0.5509 | 0.4896 | 0.4847 | 0.4847 | 0.4888 | **0.6461** |
| NCI-H23 | AUROC | 0.6950 | 0.7416 | 0.7703 | 0.7284 | 0.7812 | 0.5150 | 0.5254 | 0.7656 | **0.7900** |
| | AUPRC | 0.1064 | 0.1976 | 0.1904 | 0.1299 | 0.2056 | 0.0945 | 0.0546 | 0.1916 | **0.2927** |
| | Recall@k | 0.1296 | 0.2623 | 0.2502 | 0.1595 | 0.2632 | 0.1441 | 0.0583 | 0.2421 | **0.3417** |
| | F1-score | 0.4939 | 0.5667 | 0.5979 | 0.5452 | 0.5566 | 0.4869 | 0.4869 | 0.5056 | **0.6546** |
| OVCAR-8 | AUROC | 0.6791 | 0.7464 | 0.7296 | 0.6917 | 0.7467 | 0.5213 | 0.5148 | 0.7152 | **0.7762** |
| | AUPRC | 0.0947 | 0.1797 | 0.1840 | 0.1100 | 0.2066 | 0.0834 | 0.0530 | 0.1429 | **0.2888** |
| | Recall@k | 0.1162 | 0.2452 | 0.2652 | 0.1346 | 0.2388 | 0.1474 | 0.0585 | 0.1867 | **0.3438** |
| | F1-score | 0.4882 | 0.5728 | 0.5886 | 0.5598 | 0.4893 | 0.4868 | 0.4868 | 0.4876 | **0.6461** |
| P388 | AUROC | 0.6444 | 0.7424 | 0.7269 | 0.7391 | 0.7148 | 0.6196 | 0.5169 | 0.7375 | **0.7896** |
| | AUPRC | 0.0911 | 0.2151 | 0.2413 | 0.2131 | 0.1508 | 0.2493 | 0.0615 | 0.2175 | **0.3929** |
| | Recall@k | 0.1255 | 0.3067 | 0.3336 | 0.3009 | 0.2045 | 0.2843 | 0.0718 | 0.2980 | **0.4431** |
| | F1-score | 0.4912 | 0.4942 | 0.4990 | 0.6053 | 0.4858 | 0.4858 | 0.4858 | 0.5608 | **0.7061** |

Table 9: Average performance with multiple runs (heterogeneous graph classification models).

| Datasets | Metrics | HMGNN | muxGNN | HeGCL | RFAGNN | SHGLNN | JacobiGAD |
|---|---|---|---|---|---|---|---|
| MCF-7 | AUROC | 0.3652 | 0.5570 | 0.6733 | 0.6990 | 0.5079 | **0.7679** |
| | AUPRC | 0.0646 | 0.1402 | 0.1735 | 0.1829 | 0.0774 | **0.3403** |
| | Recall@k | 0.0792 | 0.2070 | 0.2367 | 0.2186 | 0.0378 | **0.3769** |
| | F1-score | 0.4965 | 0.4848 | 0.4817 | 0.5022 | 0.4785 | **0.6597** |
| MOLT-4 | AUROC | 0.5068 | 0.5009 | 0.6675 | 0.6540 | 0.4980 | **0.7381** |
| | AUPRC | 0.1038 | 0.0809 | 0.1462 | 0.1472 | 0.0737 | **0.3097** |
| | Recall@k | 0.1242 | 0.0801 | 0.1874 | 0.1773 | 0.0483 | **0.3519** |
| | F1-score | 0.5084 | 0.5003 | 0.4794 | 0.4849 | 0.4794 | **0.6507** |
| PC-3 | AUROC | 0.5359 | 0.4511 | 0.6913 | 0.6923 | 0.5201 | **0.7677** |
| | AUPRC | 0.0903 | 0.0516 | 0.1372 | 0.1157 | 0.0547 | **0.3064** |
| | Recall@k | 0.1010 | 0.0308 | 0.1690 | 0.1456 | 0.0202 | **0.3603** |
| | F1-score | 0.4869 | 0.4909 | 0.4852 | 0.4877 | 0.4853 | **0.6394** |
| SW-620 | AUROC | 0.5392 | 0.4823 | 0.6610 | 0.6633 | 0.5012 | **0.7728** |
| | AUPRC | 0.0678 | 0.0629 | 0.1419 | 0.1114 | 0.0555 | **0.2697** |
| | Recall@k | 0.0816 | 0.0781 | 0.1978 | 0.1660 | 0.0207 | **0.3347** |
| | F1-score | 0.4854 | 0.4860 | 0.4853 | 0.4846 | 0.4847 | **0.6461** |
| NCI-H23 | AUROC | 0.3299 | 0.5302 | 0.7090 | 0.6834 | 0.5212 | **0.7900** |
| | AUPRC | 0.0357 | 0.0615 | 0.1621 | 0.1024 | 0.0490 | **0.2927** |
| | Recall@k | 0.0300 | 0.0858 | 0.2340 | 0.1482 | 0.0178 | **0.3417** |
| | F1-score | 0.4867 | 0.5160 | 0.4891 | 0.4869 | 0.4869 | **0.6546** |
| OVCAR-8 | AUROC | 0.4711 | 0.4449 | 0.6673 | 0.6691 | 0.5154 | **0.7762** |
| | AUPRC | 0.0634 | 0.0457 | 0.1188 | 0.0950 | 0.0489 | **0.2888** |
| | Recall@k | 0.0817 | 0.0481 | 0.1707 | 0.1282 | 0.0224 | **0.3438** |
| | F1-score | 0.4867 | 0.4872 | 0.4883 | 0.4960 | 0.4868 | **0.6461** |
| P388 | AUROC | 0.5734 | 0.4296 | 0.5765 | 0.7270 | 0.6333 | **0.7896** |
| | AUPRC | 0.0925 | 0.0632 | 0.0749 | 0.1745 | 0.0774 | **0.3929** |
| | Recall@k | 0.1313 | 0.1066 | 0.0964 | 0.2379 | 0.0718 | **0.4431** |
| | F1-score | 0.4858 | 0.5284 | 0.4878 | 0.5663 | 0.4858 | **0.7061** |

Table 9, and graph-level anomaly detection methods, shown in Table 10. These results reinforce the effectiveness and broad applicability of our approach and confirm that the improvements are not confined to a narrow set of benchmarks but hold across a diverse collection of graph domains.

# F ABLATION STUDY

In this section, we will further analyze the influence of different components in JacobiGAD. To be specific, we will investigate different components in three dimensions, that is, component deactivation (additional experiments), input replacement, and polynomial degradation.

Table 10: Average performance with multiple runs (GAD models).

| Datasets | Metrics | iGAD | GmapAD | RumorMixer | RQGNN | UniGAD | JacobiGAD |
|---|---|---|---|---|---|---|---|
| MCF-7 | AUROC | 0.7140 | 0.5889 | 0.3951 | 0.7332 | 0.5987 | **0.7679** |
| | AUPRC | 0.1913 | 0.1001 | 0.0663 | 0.2585 | 0.1133 | **0.3403** |
| | Recall@k | 0.2629 | 0.1147 | 0.0621 | 0.3065 | 0.1467 | **0.3769** |
| | F1-score | 0.5637 | 0.4046 | 0.4785 | 0.5768 | 0.5091 | **0.6597** |
| MOLT-4 | AUROC | 0.7111 | 0.6108 | 0.4985 | 0.7082 | 0.5880 | **0.7381** |
| | AUPRC | 0.2025 | 0.1018 | 0.0789 | 0.2248 | 0.1067 | **0.3097** |
| | Recall@k | 0.2749 | 0.1056 | 0.0786 | 0.2845 | 0.1433 | **0.3519** |
| | F1-score | 0.5766 | 0.4351 | 0.4794 | 0.6072 | 0.4915 | **0.6507** |
| PC-3 | AUROC | 0.7040 | 0.5707 | 0.3878 | 0.7260 | 0.6308 | **0.7677** |
| | AUPRC | 0.1254 | 0.0665 | 0.0448 | 0.2143 | 0.0958 | **0.3064** |
| | Recall@k | 0.1807 | 0.0755 | 0.0287 | 0.2880 | 0.1403 | **0.3603** |
| | F1-score | 0.5117 | 0.3826 | 0.4853 | 0.6207 | 0.5091 | **0.6394** |
| SW-620 | AUROC | 0.7280 | 0.6058 | 0.4230 | 0.7687 | 0.6195 | **0.7728** |
| | AUPRC | 0.1699 | 0.0760 | 0.0498 | 0.2105 | 0.0972 | **0.2697** |
| | Recall@k | 0.2254 | 0.0761 | 0.0346 | 0.2621 | 0.1480 | **0.3347** |
| | F1-score | 0.5606 | 0.4092 | 0.4847 | 0.5883 | 0.5000 | **0.6461** |
| NCI-H23 | AUROC | 0.7531 | 0.5556 | 0.4007 | 0.7817 | 0.6276 | **0.7900** |
| | AUPRC | 0.1616 | 0.0572 | 0.0408 | 0.2618 | 0.0877 | **0.2927** |
| | Recall@k | 0.2316 | 0.0615 | 0.0324 | 0.3142 | 0.1377 | **0.3417** |
| | F1-score | 0.5577 | 0.2746 | 0.4869 | 0.6258 | 0.5087 | **0.6546** |
| OVCAR-8 | AUROC | 0.7205 | 0.6147 | 0.4042 | 0.7381 | 0.5975 | **0.7762** |
| | AUPRC | 0.1449 | 0.0673 | 0.0414 | 0.1973 | 0.0734 | **0.2888** |
| | Recall@k | 0.2228 | 0.0793 | 0.0248 | 0.2596 | 0.0954 | **0.3438** |
| | F1-score | 0.5538 | 0.3971 | 0.4868 | 0.5903 | 0.4542 | **0.6461** |
| P388 | AUROC | 0.6776 | 0.5620 | 0.4369 | 0.7625 | 0.6065 | **0.7896** |
| | AUPRC | 0.1989 | 0.0643 | 0.0504 | 0.2572 | 0.0763 | **0.3929** |
| | Recall@k | 0.2828 | 0.0790 | 0.0464 | 0.3256 | 0.0892 | **0.4431** |
| | F1-score | 0.5622 | 0.4655 | 0.4858 | 0.5559 | 0.4862 | **0.7061** |

Table 11: Ablation study for component deactivation.

| Datasets | Metrics | JacobiGAD | w/o $\mathcal{L}_{RFACE}$ | w/o learnable $(\alpha, \beta)$ | w/o learnable $\omega_r$ |
|---|---|---|---|---|---|
| MCF-7 | AUROC | 0.7679 | 0.7400 | 0.7358 | 0.7181 |
| | AUPRC | 0.3403 | 0.2794 | 0.2763 | 0.2517 |
| | Recall@k | 0.3769 | 0.3232 | 0.3275 | 0.2992 |
| | F1-score | 0.6597 | 0.5945 | 0.5982 | 0.6124 |
| MOLT-4 | AUROC | 0.7381 | 0.7216 | 0.7280 | 0.7186 |
| | AUPRC | 0.3097 | 0.2927 | 0.2593 | 0.2807 |
| | Recall@k | 0.3519 | 0.3455 | 0.3174 | 0.3349 |
| | F1-score | 0.6507 | 0.6375 | 0.6302 | 0.6400 |
| PC-3 | AUROC | 0.7677 | 0.7526 | 0.7308 | 0.7584 |
| | AUPRC | 0.3064 | 0.2589 | 0.2578 | 0.2780 |
| | Recall@k | 0.3603 | 0.2986 | 0.3209 | 0.3390 |
| | F1-score | 0.6394 | 0.6087 | 0.6172 | 0.6378 |
| SW-620 | AUROC | 0.7728 | 0.7457 | 0.7480 | 0.7553 |
| | AUPRC | 0.2697 | 0.2434 | 0.2467 | 0.2568 |
| | Recall@k | 0.3347 | 0.3071 | 0.3119 | 0.3098 |
| | F1-score | 0.6461 | 0.6031 | 0.6210 | 0.6236 |
| NCI-H23 | AUROC | 0.7900 | 0.7684 | 0.7895 | 0.7735 |
| | AUPRC | 0.2927 | 0.2444 | 0.2647 | 0.2899 |
| | Recall@k | 0.3417 | 0.3158 | 0.3142 | 0.3409 |
| | F1-score | 0.6546 | 0.5967 | 0.6395 | 0.6265 |
| OVCAR-8 | AUROC | 0.7762 | 0.7521 | 0.7636 | 0.7578 |
| | AUPRC | 0.2888 | 0.2189 | 0.2532 | 0.2371 |
| | Recall@k | 0.3438 | 0.2716 | 0.3117 | 0.2925 |
| | F1-score | 0.6461 | 0.6003 | 0.6130 | 0.6022 |
| P388 | AUROC | 0.7896 | 0.7332 | 0.7699 | 0.7733 |
| | AUPRC | 0.3929 | 0.3262 | 0.3693 | 0.3729 |
| | Recall@k | 0.4431 | 0.3836 | 0.4032 | 0.4119 |
| | F1-score | 0.7061 | 0.6762 | 0.6354 | 0.6698 |

## F.1 Additional Component Deactivation

To further validate the contribution of each component in our framework, we conduct an extended ablation study on 7 other benchmark datasets: MCF-7, MOLT-4, PC-3, SW-620, NCI-H23, OVCAR-8, and P388.

In this expanded evaluation of Table 11, we follow the same setting shown in Section 5.3. Across all datasets, the full model consistently achieves the highest detection scores, while removing any major component leads to clear and reproducible degradation. These results collectively demonstrate

Table 12: Ablation study for input replacement.

| Datasets | Metrics | JacobiGAD | SVD | Concat | MLP |
|---|---|---|---|---|---|
| MCF-7 | AUROC | 0.7679 | 0.7258 | 0.7235 | 0.7393 |
| | AUPRC | 0.3403 | 0.2549 | 0.2459 | 0.3060 |
| | Recall@k | 0.3769 | 0.2905 | 0.3028 | 0.3508 |
| | F1-score | 0.6597 | 0.5935 | 0.5972 | 0.6419 |
| MOLT-4 | AUROC | 0.7381 | 0.7257 | 0.7223 | 0.7106 |
| | AUPRC | 0.3097 | 0.2607 | 0.2745 | 0.2412 |
| | Recall@k | 0.3519 | 0.3137 | 0.3471 | 0.3052 |
| | F1-score | 0.6507 | 0.6292 | 0.6377 | 0.6163 |
| PC-3 | AUROC | 0.7677 | 0.7645 | 0.7378 | 0.7397 |
| | AUPRC | 0.3064 | 0.2669 | 0.2450 | 0.2537 |
| | Recall@k | 0.3603 | 0.3220 | 0.2880 | 0.3092 |
| | F1-score | 0.6394 | 0.5849 | 0.6113 | 0.6256 |
| SW-620 | AUROC | 0.7728 | 0.7576 | 0.7322 | 0.7365 |
| | AUPRC | 0.2697 | 0.2508 | 0.2233 | 0.2319 |
| | Recall@k | 0.3347 | 0.3133 | 0.2863 | 0.2953 |
| | F1-score | 0.6461 | 0.6209 | 0.6162 | 0.6195 |
| NCI-H23 | AUROC | 0.7900 | 0.7718 | 0.7319 | 0.7474 |
| | AUPRC | 0.2927 | 0.2556 | 0.2317 | 0.2394 |
| | Recall@k | 0.3417 | 0.3320 | 0.2858 | 0.2947 |
| | F1-score | 0.6546 | 0.6304 | 0.6046 | 0.6129 |
| OVCAR-8 | AUROC | 0.7762 | 0.7567 | 0.7444 | 0.7729 |
| | AUPRC | 0.2888 | 0.2333 | 0.2319 | 0.2462 |
| | Recall@k | 0.3438 | 0.2877 | 0.2965 | 0.2901 |
| | F1-score | 0.6461 | 0.5892 | 0.5923 | 0.5889 |
| P388 | AUROC | 0.7896 | 0.7779 | 0.7862 | 0.7439 |
| | AUPRC | 0.3929 | 0.3501 | 0.3702 | 0.3376 |
| | Recall@k | 0.4431 | 0.4054 | 0.4271 | 0.4054 |
| | F1-score | 0.7061 | 0.6592 | 0.6938 | 0.6760 |
| SF-295 | AUROC | 0.7729 | 0.7554 | 0.7563 | 0.7421 |
| | AUPRC | 0.2623 | 0.2179 | 0.1935 | 0.1820 |
| | Recall@k | 0.3210 | 0.3004 | 0.2733 | 0.2502 |
| | F1-score | 0.6356 | 0.6124 | 0.6103 | 0.5859 |
| SN12C | AUROC | 0.7797 | 0.7682 | 0.7523 | 0.7568 |
| | AUPRC | 0.2666 | 0.2457 | 0.2671 | 0.2430 |
| | Recall@k | 0.3240 | 0.3078 | 0.3291 | 0.3018 |
| | F1-score | 0.6329 | 0.6329 | 0.6380 | 0.6221 |
| UACC257 | AUROC | 0.7613 | 0.7692 | 0.7440 | 0.7505 |
| | AUPRC | 0.1995 | 0.2218 | 0.1660 | 0.1988 |
| | Recall@k | 0.2819 | 0.2901 | 0.2475 | 0.2677 |
| | F1-score | 0.6246 | 0.6214 | 0.5775 | 0.5930 |
| DBLP | AUROC | 0.9830 | 0.9679 | 0.9780 | 0.9805 |
| | AUPRC | 0.9842 | 0.9665 | 0.9796 | 0.9822 |
| | Recall@k | 0.9575 | 0.9374 | 0.9441 | 0.9508 |
| | F1-score | 0.9651 | 0.9294 | 0.9576 | 0.9623 |
| IMDB | AUROC | 0.7263 | 0.6665 | 0.6961 | 0.6905 |
| | AUPRC | 0.7619 | 0.7282 | 0.7318 | 0.7215 |
| | Recall@k | 0.7192 | 0.6824 | 0.7035 | 0.6951 |
| | F1-score | 0.6585 | 0.6046 | 0.6445 | 0.6402 |
| PDNS | AUROC | 0.8728 | 0.8697 | 0.8673 | 0.8707 |
| | AUPRC | 0.6871 | 0.6650 | 0.6801 | 0.6865 |
| | Recall@k | 0.6283 | 0.6206 | 0.6204 | 0.6247 |
| | F1-score | 0.7760 | 0.7526 | 0.7708 | 0.7676 |
| RCDD | AUROC | 0.9826 | 0.9777 | 0.9829 | 0.9806 |
| | AUPRC | 0.9332 | 0.9174 | 0.9322 | 0.9288 |
| | Recall@k | 0.8747 | 0.8550 | 0.8775 | 0.8623 |
| | F1-score | 0.9280 | 0.9161 | 0.9284 | 0.9194 |

that each component contributes meaningfully to the final performance and that their combination is essential for achieving the strong detection capability of our method.

## F.2 INPUT REPLACEMENT

Next, we investigate the influence of different ways of input for JacobiGAD, i.e., SVD, Concat, and MLP.

As shown in Table 12, this ablation study evaluates the efficacy of the proposed input function in JacobiGAD for unifying features from different views in a heterogeneous graph by comparing it

against three common alternative methods: SVD (which may lose critical information), Concat (which creates a high-dimensional feature space), and MLP (which causes higher computational cost and may easily overfit). The results consistently demonstrate that JacobiGAD's specialized integration method, Gaussian projection, significantly outperforms all three alternatives across the vast majority of datasets and metrics. Although there are rare, minor exceptions where an alternative method performs comparably on some datasets, the overall trend is unequivocal: the custom-designed input function in JacobiGAD is uniquely capable of effectively synthesizing heterogeneous information, which is a critical factor in the model's superior anomaly detection performance.

### F.3 Polynomial Degradation

Finally, we analyze the influence of different polynomials for JacobiGAD, i.e., Gegenbauer ($\alpha = \beta = \lambda - \frac{1}{2}$), Chebyshev ($\alpha = \beta = -\frac{1}{2}$), and Legendre ($\alpha = \beta = 0$).

As shown in Table 13, the ablation study demonstrates that the choice of polynomial basis for the graph filter is critical, with the proposed Jacobi polynomials consistently outperforming Gegenbauer, Chebyshev, and Legendre polynomials across all datasets and metrics. The key drawback of these alternative polynomials is their inherent rigidity. Unlike the parameter-rich Jacobi basis, which can be adaptively tuned to fit the complex spectral characteristics of heterogeneous graphs, the fixed spectral response of Chebyshev and Legendre polynomials and the limited single-parameter flexibility of Gegenbauer polynomials render them less capable of capturing the nuanced patterns necessary for effective anomaly detection. This lack of adaptability manifests clearly in the significant performance gaps, indicating that the alternative filters struggle to generate the highly discriminative representations needed to reliably separate anomalies from normal nodes in complex graph data.

## G Learned Parameters

In this section, we will present the learned parameters of one run of our experiment to show the influence of different parameters on all datasets.

The blank slot of Table 14 is due to the best $K$ for different datasets not being the same. As shown in Table 14, the results further demonstrate the importance of learnable ($\alpha, \beta$) as the best performance of different datasets requires distinct combinations of ($\alpha, \beta$), instead of fixed parameters.

In Table 15, we present the statistical information of $\omega_r$, due to the large number of different relations in heterogeneous datasets. We use the row Range as the start and the end of the range. For example, for the first range of MCF-7, it is formed by $[-1.4456, -1.0144)$. And the corresponding frequency is reported in the row Frequency. In this case, the frequency of $[-1.4456, -1.0144)$ is 1. Other cases can be deduced by analogy. We can be informed by Table 15 that the learnable $\omega_r$ is of vital importance for heterogeneous GAD, as the best $\omega_r$ for different datasets can distribute evenly, focus on the center part, or lie mainly on the extreme spots.

## H Comparison with Focal Loss

We further compare our proposed RFACE with Focal loss, a classical loss for imbalanced data, to demonstrate the effectiveness of our proposed methods.

Assume we have logits $\boldsymbol{z} = [z_1, \cdots, z_C]$, where $C$ is the number of classes, sigmoid per class $\boldsymbol{p} = [p_1, \cdots, p_C]$, where $p_i = \text{Sigmoid}(z_i)$, and multi-label target $\boldsymbol{y} = [y_1, \cdots, y_C] \in \{0,1\}^C$, then we will investigate the gradients of Cross-Entropy loss, Focal loss, and RFACE to show the key advantages of RFACE.

For Cross-Entropy loss:

$$\mathcal{L}_{CE} = -\sum_{i=1}^{C} [y_i \log p_i + (1 - y_i) \log(1 - p_i)],$$

the gradient vector is:

$$\nabla_{\boldsymbol{z}} \mathcal{L}_{CE} = \boldsymbol{p} - \boldsymbol{y}$$

Table 13: Ablation study for polynomial degradation.

| Datasets | Metrics | JacobiGAD | Gegenbauer | Chebyshev | Legendre |
|----------|---------|-----------|------------|-----------|----------|
| MCF-7 | AUROC | 0.7679 | 0.7281 | 0.7146 | 0.6986 |
| | AUPRC | 0.3403 | 0.2787 | 0.2375 | 0.2384 |
| | Recall@k | 0.3769 | 0.3217 | 0.2847 | 0.2767 |
| | F1-score | 0.6597 | 0.6048 | 0.5889 | 0.5886 |
| MOLT-4 | AUROC | 0.7381 | 0.7180 | 0.7076 | 0.7202 |
| | AUPRC | 0.3097 | 0.2587 | 0.2476 | 0.2475 |
| | Recall@k | 0.3519 | 0.3068 | 0.3100 | 0.3132 |
| | F1-score | 0.6507 | 0.6225 | 0.6248 | 0.6224 |
| PC-3 | AUROC | 0.7677 | 0.7226 | 0.7589 | 0.7384 |
| | AUPRC | 0.3064 | 0.2057 | 0.2494 | 0.2281 |
| | Recall@k | 0.3603 | 0.2508 | 0.3092 | 0.2944 |
| | F1-score | 0.6394 | 0.5689 | 0.6061 | 0.5974 |
| SW-620 | AUROC | 0.7728 | 0.7386 | 0.7441 | 0.7381 |
| | AUPRC | 0.2697 | 0.2392 | 0.2461 | 0.2233 |
| | Recall@k | 0.3347 | 0.3105 | 0.2988 | 0.2766 |
| | F1-score | 0.6461 | 0.5995 | 0.6294 | 0.5905 |
| NCI-H23 | AUROC | 0.7900 | 0.7758 | 0.7891 | 0.7727 |
| | AUPRC | 0.2927 | 0.2413 | 0.2556 | 0.2274 |
| | Recall@k | 0.3417 | 0.3045 | 0.3296 | 0.3020 |
| | F1-score | 0.6546 | 0.5775 | 0.5679 | 0.6101 |
| OVCAR-8 | AUROC | 0.7762 | 0.7703 | 0.7733 | 0.7691 |
| | AUPRC | 0.2888 | 0.2551 | 0.2388 | 0.2367 |
| | Recall@k | 0.3438 | 0.3101 | 0.3117 | 0.3117 |
| | F1-score | 0.6461 | 0.6286 | 0.6048 | 0.6003 |
| P388 | AUROC | 0.7896 | 0.7554 | 0.7564 | 0.7656 |
| | AUPRC | 0.3929 | 0.3465 | 0.3424 | 0.3587 |
| | Recall@k | 0.4431 | 0.3952 | 0.4054 | 0.3988 |
| | F1-score | 0.7061 | 0.6855 | 0.6886 | 0.6790 |
| SF-295 | AUROC | 0.7729 | 0.7670 | 0.7578 | 0.7461 |
| | AUPRC | 0.2623 | 0.2085 | 0.1939 | 0.1919 |
| | Recall@k | 0.3210 | 0.2724 | 0.2634 | 0.2634 |
| | F1-score | 0.6356 | 0.5965 | 0.5965 | 0.6020 |
| SN12C | AUROC | 0.7797 | 0.7459 | 0.7384 | 0.7404 |
| | AUPRC | 0.2666 | 0.2284 | 0.2308 | 0.2224 |
| | Recall@k | 0.3240 | 0.3018 | 0.2864 | 0.2805 |
| | F1-score | 0.6329 | 0.6052 | 0.6080 | 0.6138 |
| UACC257 | AUROC | 0.7613 | 0.7484 | 0.7390 | 0.6997 |
| | AUPRC | 0.1995 | 0.1995 | 0.1715 | 0.1587 |
| | Recall@k | 0.2819 | 0.2708 | 0.2525 | 0.2231 |
| | F1-score | 0.6246 | 0.5656 | 0.5659 | 0.5827 |
| DBLP | AUROC | 0.9830 | 0.9778 | 0.9756 | 0.9746 |
| | AUPRC | 0.9842 | 0.9802 | 0.9750 | 0.9732 |
| | Recall@k | 0.9575 | 0.9508 | 0.9463 | 0.9418 |
| | F1-score | 0.9651 | 0.9632 | 0.9557 | 0.9539 |
| IMDB | AUROC | 0.7263 | 0.7096 | 0.7130 | 0.7060 |
| | AUPRC | 0.7619 | 0.7485 | 0.7552 | 0.7444 |
| | Recall@k | 0.7192 | 0.7119 | 0.7108 | 0.7098 |
| | F1-score | 0.6585 | 0.6527 | 0.6461 | 0.6565 |
| PDNS | AUROC | 0.8728 | 0.8724 | 0.8721 | 0.8716 |
| | AUPRC | 0.6871 | 0.6732 | 0.6844 | 0.6860 |
| | Recall@k | 0.6283 | 0.6145 | 0.6270 | 0.6241 |
| | F1-score | 0.7760 | 0.7700 | 0.7704 | 0.7702 |
| RCDD | AUROC | 0.9826 | 0.9809 | 0.9805 | 0.9815 |
| | AUPRC | 0.9332 | 0.9283 | 0.9290 | 0.9299 |
| | Recall@k | 0.8747 | 0.8741 | 0.8719 | 0.8667 |
| | F1-score | 0.9280 | 0.9279 | 0.9229 | 0.9220 |

For Focal loss:

$$\mathcal{L}_{Focal} = -\sum_{i=1}^{C}[y_i(1 - p_i)^\gamma \log p_i + (1 - y_i)p_i^\gamma \log(1 - p_i)],$$

the gradient vector is:

$$\nabla_z \mathcal{L}_{Focal} = s_{Focal}(p - y),$$

where $s_{Focal}$ is a scalar vector for each class $i$, depending on the ground truth label $y_i$, the predicted probability with no midification $p_i$, and the power for measuring the difficulty of samples $\gamma$.

Table 14: Learned $(\alpha, \beta)$ for Jacobi Polynomials.

| Datasets | $\alpha$ | | | | $\beta$ | | | |
|---|---|---|---|---|---|---|---|---|
| MCF-7 | 0.1123 | 1.2406 | 1.2030 | | 1.7027 | 1.5336 | 1.8209 | |
| MOLT-4 | 1.9611 | 0.8162 | 0.2852 | 1.8448 | 0.3078 | 1.0801 | 0.7354 | 1.5658 |
| PC-3 | 0.8800 | 1.6410 | | | 0.0865 | 0.8657 | | |
| SW-620 | 0.4661 | 1.608 | 1.3252 | 0.0181 | 1.9602 | 0.3216 | 0.4713 | 0.5893 |
| NCI-H23 | 0.3489 | 1.5247 | 1.9206 | | 1.8042 | 1.0248 | 1.3645 | |
| OVCAR-8 | 1.3458 | 1.6198 | 0.6928 | 1.0384 | 0.1690 | 0.0049 | 0.2966 | 1.1328 |
| P388 | 1.0701 | 0.5101 | 0.1645 | 0.2895 | 1.9218 | 1.5771 | 0.1775 | 0.1930 |
| SF-295 | 0.7552 | 0.326 | 1.4094 | | 0.3687 | 0.4355 | 1.8797 | |
| SN12C | 0.7820 | 0.8886 | 0.9901 | | 0.9222 | 0.1497 | 0.3569 | |
| UACC257 | 1.7794 | 1.7520 | 1.7412 | 1.9050 | 1.0449 | 0.2968 | 1.0761 | 0.3197 |
| DBLP | 1.9142 | 0.4518 | | | 1.7499 | 0.3554 | | |
| IMDB | 0.6390 | 0.5686 | | | 0.7820 | 1.3118 | | |
| PDNS | 0.6258 | 0.5431 | 1.9307 | | 0.1602 | 1.1881 | 0.7147 | |
| RCDD | 0.4016 | | | | 0.8339 | | | |

Table 15: Learned $\omega_r$ for different relations.

| Datasets | Metrics | | | | | | | | | | | |
|---|---|---|---|---|---|---|---|---|---|---|---|---|
| MCF-7 | Range | -1.4456 | -1.0144 | -0.5832 | -0.1519 | 0.2793 | 0.7105 | 1.1417 | 1.5729 | 2.0042 | 2.4354 | 2.8666 |
| | Frequency | 1 | 0 | 1 | 19 | 30 | 27 | 16 | 25 | 7 | 3 | |
| MOLT-4 | Range | 0.0200 | 0.2136 | 0.4071 | 0.6006 | 0.7942 | 0.9877 | 1.1813 | 1.3748 | 1.5684 | 1.7619 | 1.9555 |
| | Frequency | 19 | 12 | 12 | 25 | 22 | 18 | 19 | 17 | 18 | 14 | |
| PC-3 | Range | 0.0103 | 0.2200 | 0.4297 | 0.6394 | 0.8490 | 1.0587 | 1.2684 | 1.4781 | 1.6878 | 1.8974 | 2.1071 |
| | Frequency | 14 | 10 | 19 | 14 | 14 | 10 | 14 | 8 | 30 | 10 | |
| SW-620 | Range | -1.1965 | -0.6950 | -0.1936 | 0.3078 | 0.8093 | 1.3107 | 1.8121 | 2.3136 | 2.8150 | 3.3164 | 3.8179 |
| | Frequency | 2 | 3 | 20 | 44 | 45 | 42 | 24 | 2 | 1 | 1 | |
| NCI-H23 | Range | -0.0007 | 0.2072 | 0.4151 | 0.6230 | 0.8309 | 1.0389 | 1.2468 | 1.4547 | 1.6626 | 1.8705 | 2.0784 |
| | Frequency | 17 | 21 | 14 | 21 | 17 | 18 | 21 | 14 | 24 | 15 | |
| OVCAR-8 | Range | -0.0148 | 0.1907 | 0.3963 | 0.6019 | 0.8074 | 1.0130 | 1.2185 | 1.4241 | 1.6296 | 1.8352 | 2.0407 |
| | Frequency | 22 | 16 | 21 | 18 | 22 | 15 | 13 | 19 | 17 | 21 | |
| P388 | Range | 0.0009 | 0.2002 | 0.3995 | 0.5988 | 0.7981 | 0.9974 | 1.1967 | 1.3960 | 1.5953 | 1.7946 | 1.9939 |
| | Frequency | 24 | 30 | 26 | 26 | 19 | 28 | 31 | 27 | 26 | 34 | |
| SF-295 | Range | -0.1725 | 0.0563 | 0.2852 | 0.5140 | 0.7429 | 0.9717 | 1.2006 | 1.4294 | 1.6583 | 1.8871 | 2.1160 |
| | Frequency | 8 | 22 | 24 | 26 | 19 | 22 | 13 | 17 | 22 | 11 | |
| SN12C | Range | -0.0906 | 0.1206 | 0.3317 | 0.5428 | 0.7540 | 0.9651 | 1.1762 | 1.3873 | 1.5985 | 1.8096 | 2.0207 |
| | Frequency | 12 | 17 | 28 | 22 | 20 | 21 | 19 | 18 | 17 | 10 | |
| UACC257 | Range | -0.0503 | 0.1549 | 0.3601 | 0.5653 | 0.7705 | 0.9757 | 1.1809 | 1.3861 | 1.5913 | 1.7965 | 2.0017 |
| | Frequency | 8 | 10 | 18 | 19 | 29 | 10 | 16 | 22 | 23 | 21 | |
| DBLP | Range | -0.0094 | 0.1563 | 0.3220 | 0.4878 | 0.6535 | 0.8193 | 0.9850 | 1.1507 | 1.3165 | 1.4822 | 1.6480 |
| | Frequency | 2 | 0 | 0 | 0 | 1 | 0 | 1 | 1 | 0 | 1 | |
| IMDB | Range | -0.0099 | 0.1620 | 0.3339 | 0.5058 | 0.6777 | 0.8496 | 1.0214 | 1.1933 | 1.3652 | 1.5371 | 1.7090 |
| | Frequency | 1 | 0 | 0 | 1 | 0 | 0 | 0 | 0 | 0 | 2 | |
| PDNS | Range | 0.3526 | 0.4796 | 0.6065 | 0.7335 | 0.8604 | 0.9874 | 1.1143 | 1.2413 | 1.3682 | 1.4952 | 1.6221 |
| | Frequency | 1 | 0 | 0 | 0 | 1 | 0 | 0 | 0 | 0 | 1 | |
| RCDD | Range | 0.7637 | 0.8787 | 0.9938 | 1.1089 | 1.2240 | 1.3390 | 1.4541 | 1.5692 | 1.6842 | 1.7993 | 1.9144 |
| | Frequency | 3 | 0 | 1 | 1 | 0 | 1 | 0 | 0 | 0 | 2 | |

For RFACE:

$$\mathcal{L}_{RFACE} = -\sum_{i=1}^{C}[y_i \log p'_i + (1 - y_i) \log(1 - p'_i)],$$

$$p'_i = \text{Sigmoid}(z_i + \gamma \kappa_i \nabla_{z_i} \mathcal{L}_{CE}),$$

the gradient vector is:

$$\nabla_{\boldsymbol{z}} \mathcal{L}_{RFACE} = \boldsymbol{s}_{RFACE}(\boldsymbol{p}' - \boldsymbol{y}),$$

where $\boldsymbol{s}_{RFACE}$ is a scalar vector for each class $i$, depending on the ground truth label $y_i$, the predicted probability with logit modification (indicating class frequency by $\kappa_i$ and sample difficulty by $\nabla_{z_i} \mathcal{L}_{CE}$) $p_i$, the coefficient of modification term $\gamma$, and the class-frequency-based curvature $\kappa_i$.

As shown above, the advantage of the proposed RFACE over Focal loss stems from its distinct mechanism for addressing class imbalance in heterogeneous graph-level anomaly detection. RFACE

Table 16: Ablation study for Focal Loss.

| Datasets | Metrics | JacobiGAD | w/ Focal Loss |
|---|---|---|---|
| MCF-7 | AUROC | 0.7679 | 0.7544 |
| | AUPRC | 0.3403 | 0.2846 |
| | Recall@k | 0.3769 | 0.3261 |
| | Macro-F1 | 0.6597 | 0.6106 |
| MOLT-4 | AUROC | 0.7381 | 0.7273 |
| | AUPRC | 0.3097 | 0.2612 |
| | Recall@k | 0.3519 | 0.3429 |
| | Macro-F1 | 0.6507 | 0.5796 |
| PC-3 | AUROC | 0.7677 | 0.7490 |
| | AUPRC | 0.3064 | 0.2275 |
| | Recall@k | 0.3603 | 0.3050 |
| | Macro-F1 | 0.6394 | 0.6203 |
| SW-620 | AUROC | 0.7728 | 0.7497 |
| | AUPRC | 0.2697 | 0.2537 |
| | Recall@k | 0.3347 | 0.3264 |
| | Macro-F1 | 0.6461 | 0.6281 |
| NCI-H23 | AUROC | 0.7900 | 0.7417 |
| | AUPRC | 0.2927 | 0.2281 |
| | Recall@k | 0.3417 | 0.3028 |
| | Macro-F1 | 0.6546 | 0.6255 |
| OVCAR-8 | AUROC | 0.7762 | 0.7310 |
| | AUPRC | 0.2888 | 0.2259 |
| | Recall@k | 0.3438 | 0.2925 |
| | Macro-F1 | 0.6461 | 0.5880 |
| P388 | AUROC | 0.7896 | 0.7503 |
| | AUPRC | 0.3929 | 0.3136 |
| | Recall@k | 0.4431 | 0.3749 |
| | Macro-F1 | 0.7061 | 0.6431 |
| SF-295 | AUROC | 0.7729 | 0.7369 |
| | AUPRC | 0.2623 | 0.1916 |
| | Recall@k | 0.3210 | 0.2691 |
| | Macro-F1 | 0.6356 | 0.5812 |
| SN12C | AUROC | 0.7797 | 0.7682 |
| | AUPRC | 0.2666 | 0.2182 |
| | Recall@k | 0.3240 | 0.2975 |
| | Macro-F1 | 0.6329 | 0.5929 |
| UACC257 | AUROC | 0.7613 | 0.7558 |
| | AUPRC | 0.1995 | 0.1749 |
| | Recall@k | 0.2819 | 0.2414 |
| | Macro-F1 | 0.6246 | 0.5580 |
| DBLP | AUROC | 0.9830 | 0.9754 |
| | AUPRC | 0.9842 | 0.9647 |
| | Recall@k | 0.9575 | 0.9463 |
| | Macro-F1 | 0.9651 | 0.9550 |
| IMDB | AUROC | 0.7263 | 0.6874 |
| | AUPRC | 0.7619 | 0.7371 |
| | Recall@k | 0.7192 | 0.6940 |
| | Macro-F1 | 0.6585 | 0.6291 |
| PDNS | AUROC | 0.8728 | 0.8612 |
| | AUPRC | 0.6871 | 0.6611 |
| | Recall@k | 0.6283 | 0.5981 |
| | Macro-F1 | 0.7760 | 0.7297 |
| RCDD | AUROC | 0.9826 | 0.9797 |
| | AUPRC | 0.9332 | 0.9306 |
| | Recall@k | 0.8747 | 0.8741 |
| | Macro-F1 | 0.9280 | 0.9276 |

applies a class-dependent and difficulty-aware logit transformation. This transformation modifies the optimization gradient based on both class frequency and sample difficulty. As a result, RFACE reshapes the decision boundary by explicitly expanding the margins of minority anomaly classes and gently contracting the margins of dominant normal classes, based on both class and difficulty. This curvature-inspired adjustment acts like a discrete Ricci flow step, improving the geometric regularity of the representation space by amplifying deviations. In heterogeneous graph data, where anomalies arise from subtle irregularities and specialized node–edge interactions, such margin rebalancing is

crucial: it ensures that minority classes receive sustained and directionally beneficial updates even when the classifier becomes confident about them, preventing premature gradient vanishing.

In contrast, Focal loss only rescales the Cross-Entropy gradient through a difficulty-based factor $s$, which offers no mechanism to correct class-frequency–induced imbalance. Because Focal loss downweights "easy" samples regardless of their class, it may inadvertently suppress minority-class gradients once the model becomes moderately confident, leading to possible overfitting to "hard" samples. Moreover, Focal loss treats all classes identically and cannot incorporate global distributional information; the optimization trajectory therefore lacks the class-dependent curvature adjustment that RFACE introduces. This makes Focal loss sensitive to the randomness of minibatch composition, more prone to instability on small anomalous sets, and often ineffective when many anomalies are not "hard" samples, where class number information rather than prediction confidence determines anomaly separability.

Beyond gradient modification, RFACE also provides additional advantages, as proved in Theorems 8 (RFACE will amplify information of the minority class) and 9 (RFACE will converge), while Focal loss is a heuristic-oriented loss without guarantee. These theoretical advantages are also strongly supported by our empirical results. As shown in Table 16, RFACE consistently outperforms the Focal loss variant on all datasets.

Together, these theoretical insights and empirical observations demonstrate that RFACE is significantly better suited than Focal loss for heterogeneous graph-level anomaly detection, offering stronger geometric corrections, more stable optimization, better calibration, and improved exploitation of graph information.

