# OpenReview forum: "JacobiGAD: Jacobi Polynomial–Powered Heterogeneous Graph-Level Anomaly Detection"
_ICLR.cc/2026/Conference — Submitted to ICLR 2026_

### Official Review · Reviewer_pgtn · 2025-10-29

**Soundness:** 2
**Presentation:** 1
**Contribution:** 3
**Rating:** 2
**Confidence:** 3

**Summary:**

JacobiGAD is a framework for heterogeneous graph-level anomaly detection that integrates a Jacobi-polynomial–based graph neural network (JPGNN) with a Ricci Flow Adaptive Curvature Enhancement (RFACE) loss.

**Strengths:**

* The use of Jacobi polynomials as a learnable graph filter is new; theoretical analysis (Theorems 2–7) proves convergence, information preservation, and bounded approximation error.

* Multi-view fusion via Jacobi filters yields both injectivity and signal-to-noise amplification proportional to the number of views.

* Links Jacobi bases to Laplace–Beltrami eigenfunctions across Euclidean, spherical, and hyperbolic geometries, suggesting geometric generality.

**Weaknesses:**

* The paper is extremely difficult to follow. There is no clear narrative flow or visual guidance — the entire text consists mainly of dense equations, scattered theorems, and tables of numbers. While the method itself actually follows a relatively straightforward pipeline (JacobiGAD = Feature Alignment + Multi-view Fusion + JPGNN + RFACE Loss), the paper fails to convey this structure clearly. Including an overview figure or intuitive illustrations would make it much easier for readers to grasp. The numerous proofs and theoretical claims should be placed in the appendix rather than interrupting the main story.

* The authors claim that “most existing anomaly detection models can only handle homogeneous graphs,” which is an oversimplified and somewhat misleading statement. Ironically, their own results show that heterogeneous models underperform homogeneous ones, without offering any explanation for this phenomenon. Moreover, the related work section cites multiple heterogeneous graph classification methods, yet the paper asserts that such models cannot handle heterogeneity — this contradiction undermines the central motivation. In reality, many heterogeneous graph-level anomaly detection approaches already exist (e.g., [1–4]), and the authors should clearly position their work among them.

* The authors have not released their code, making it impossible to verify whether the reported training procedures truly match the described algorithm. Additionally, evaluation is limited to performance metrics (AUROC, AUPRC, etc.) without consideration of computational aspects such as training time, memory consumption, or scalability. For a model introducing high-order polynomial filters, such analysis is essential for a fair comparison.

[1]HRGCN: Heterogeneous Graph-level Anomaly Detection with Hierarchical Relation-augmented Graph Neural Networks.
[2]Chi-Square Wavelet Graph Neural Networks for Heterogeneous Graph Anomaly Detection.
[3]FiGraph: A Dynamic Heterogeneous Graph Dataset for Financial Anomaly Detection
[4]Deep Graph Anomaly Detection: A Survey and New Perspectives.

**Questions:**

See weaknesss

---

> ### Author Response · Authors · 2025-11-20
> **Response to Reviewer pgtn (1/2)**
>
> We appreciate your comprehensive and constructive review. Your crucial comments on experiments are exceedingly helpful for us to improve our manuscript. Our point-to-point responses to your comments are given below.
>
> ---
>
> **W1**: We thank the reviewer for the suggestions for the paper writing, and we believe an additional overview figure can help the readers to understand our framework more easily, so we will adjust our main context accordingly. However, we respectively disagree with the statement that our paper is extremely difficult to follow, as the reviewer clearly grasps the main architecture of the framework, and all the theorems are surrounded by easy-to-understand context to illustrate the key properties of each of the components, i.e., Theorem 1 for Feature Alignment, Theorems 2, 3, 4, 5, 6, and 7 for JPGNN, and Theorems 8 and 9 for RFACE. The introduction of each component is built on solid theoretical analyses, which should be part of the main story instead of an interruption. Besides, although we include 9 theoretical claims in the main context, we have placed all the proofs in Appendix A, which is a reasonable arrangement.  Furthermore, our writing follows a natural flow style similar to many papers in previous ICLR, such as [1-6], which further proves the rationale of our writing style.
>
> [1] Daniel Herbst, Stefanie Jegelka. Higher-Order Graphon Neural Networks: Approximation and Cut Distance. ICLR 2025.
>
> [2] Xiang Cheng, Lawrence Carin, Suvrit Sra. Graph Transformers Dream of Electric Flow. ICLR 2025.
>
> [3] Michael Scholkemper, Xinyi Wu, Ali Jadbabaie, Michael T. Schaub. Residual Connections and Normalization Can Provably Prevent Oversmoothing in GNNs. ICLR 2025.
>
> [4] Shih-Hsin Wang, Yung-Chang Hsu, Justin Baker, Andrea L. Bertozzi, Jack Xin, Bao Wang. Rethinking the Benefits of Steerable Features in 3D Equivariant Graph Neural Networks. ICLR 2024.
>
> [5] Thien Le, Luana Ruiz, Stefanie Jegelka. A Poincaré Inequality and Consistency Results for Signal Sampling on Large Graphs. ICLR 2024.
>
> [6] Yassine Abbahaddou, Sofiane Ennadir, Johannes F. Lutzeyer, Michalis Vazirgiannis, Henrik Boström. Bounding the Expected Robustness of Graph Neural Networks Subject to Node Feature Attacks. ICLR 2024.

---

> > ### Author Response · Authors · 2025-11-20
> > **Response to Reviewer pgtn (2/2)**
> >
> > ---
> >
> > **W2**: We respectively disagree with the comments about the misleading statement: We do not include the strong statement “most existing anomaly detection models can only handle homogeneous graphs” in our original manuscript. Therefore, we kindly request the reviewer to point out where the statement exists in our original manuscript, to see what causes the misunderstanding, so that we can further discuss the statement.
> >
> > Besides, for heterogeneous models, we have explained their possible drawbacks in Sections 2 and 5 in our original manuscript, which is the main reason they might underperform homogeneous models in some cases. We will further include additional explanations in our revised paper to illustrate the phenomenon clearly.
> >
> > We also disagree with the comments about the assertion: We do not include the strong assertion “such models cannot handle heterogeneity”. The related original texts are as follows: “Although these methods perform heterogeneous graph classification, they rely on fixed filters or heuristic fusion strategies, assume balanced data, and lack principled mechanisms for anomaly detection.” and “these methods are often limited by their reliance on suboptimal integration of heterogeneous information and their failure to tackle data imbalance.” We believe our analysis is moderate and shows the possible reasons why they perform badly for heterogeneous graph-level anomaly detection, while the reviewer overstates our claim to a large extent. Therefore, we kindly request the reviewer to point out where the statement exists in our original manuscript, to see what causes the misunderstanding.
> >
> > We also need to point out that the works cited by the reviewer are from orthogonal domains: [1] is for unsupervised heterogeneous graph-level anomaly detection, [2] is for supervised heterogeneous node-level anomaly detection, [3] is a dataset for supervised heterogeneous node-level anomaly detection without any implemented method, and [4] is a survey for a broader view of the whole graph anomaly detection area, where the only mentioned work for heterogeneous graph-level anomaly detection is [1], whereas our framework targets supervised heterogeneous graph-level anomaly detection, which is a real industrial problem. We have tried our best to search related work, yet the presented works in our original manuscript are the most representative ones. We are also willing to conduct further comparison if the reviewer can introduce several other baselines that target the same setting as our work.
> >
> > ---
> >
> > **W3**: We have already released code in supplemental materials, which is the official way to provide code. And we have also included the time analysis in Appendix C in our original manuscript. To further address the reviewer’s concern, we will include empirical running time cost and memory cost in Appendix C in our revised manuscript.
> >
> > ---
> >
> > We sincerely appreciate your time, and we are glad to answer any additional questions you may have.

---

> > > ### Comment · Reviewer_pgtn · 2025-11-21
> > >
> > > Given the author's diligent response, I will increase my score as an encouragement.
> > >
> > > Regarding writing styles, opinions vary. I speak only for myself. Additionally, concerning the statement “most existing anomaly detection models can only handle homogeneous graphs,” I did not find it in the current version. Therefore, it may be my own subjective speculation, and the author may disregard it.

---

> > > > ### Author Response · Authors · 2025-11-28
> > > > **Response to Reviewer pgtn (1/1)**
> > > >
> > > > We believe a fair and objective discussion about our contribution would help the decision process. Therefore, we will reemphasize the contribution of our work. Our paper is both novel and sound from **three main perspectives**:
> > > >
> > > > **Theoretical guarantee**: Our work is grounded in a strong and comprehensive theoretical foundation, supported by 9 theorems that justify the design of all major components. Theorem 1 proves distance preservation in our feature alignment module. Theorems 2, 3, 4, 5, 6, and 7 rigorously establish the optimality and importance of JPGNN for heterogeneous graph-level anomaly detection, demonstrating that Jacobi polynomials form an optimal basis, preserve cross-view information, enhance informative signals while suppressing noise, capture geometric information across Euclidean/Hyperbolic/Spherical spaces, and offer universal approximation guarantees even with low-degree expansions. Theorems 8 and 9 further show that our RFACE balances class weights for dealing with imbalanced data and converges under standard assumptions. This theoretical framework provides principled justification for each of our architectural decisions.
> > > >
> > > > **Solid framework**: Built on the solid theoretical foundation, our method integrates feature alignment, Jacobi-Polynomial-based message passing, and Ricci-Flow-based loss reweighting into a coherent architecture tailored for heterogeneous graph-level anomaly detection. The framework addresses heterogeneity, multi-view fusion, noise suppression, geometric complexity, and class imbalance, which are longstanding challenges in heterogeneous graph-level anomaly detection, through components that are not only well-motivated but mathematically grounded. The result is a unified, principled, and robust system rather than a collection of heuristics.
> > > >
> > > > **Outstanding performance**: Our empirical results further demonstrate the effectiveness of the proposed approach. Across all datasets and all comparison groups, including homogeneous GNNs, heterogeneous GNNs, and graph-level anomaly detection baselines, our method achieves state-of-the-art performance with consistent and substantial improvements. Ablation studies confirm the importance of each component, and additional robustness analyses validate the stability of the approach. These empirical findings align with the theoretical guarantees and highlight the practical relevance of our contributions.
> > > >
> > > > We sincerely appreciate the time and effort that the reviewers have devoted to evaluating our submission. However, we would like to respectfully express a concern regarding the nature of the reviewer’s evaluation. The current review does not engage with the technical content, theoretical contributions, methodological design, or experimental results of our work, and instead appears to rely predominantly on subjective impressions. Without discussion of the paper’s substance, it is difficult for us to understand the basis of the assigned rating or provide meaningful clarifications.
> > > >
> > > > In contrast, another reviewer, jUAp, directly engaged with our contributions, acknowledging the significance of our theoretical foundation and extensive empirical validation. Given the depth and rigor of our submission, we believe a fair assessment would benefit from engagement with the actual content of the paper.
> > > >
> > > > As an author and a reviewer of ICLR 2026, we believe that an appropriate discussion would help both the author and the reviewer sides to contribute to the research community. Besides, solely relying on SPC, PC, and AC to make the final decision without providing enough discussion context would be unprofessional as a reviewer for such a top-tier conference, and would cause a huge burden on the research community.
> > > >
> > > > We fully respect the diversity of reviewer perspectives and welcome all constructive criticism. Our intention here is not to dispute differing opinions, but to request that evaluations be grounded in the content of the submission. We would be grateful if the reviewer could clarify the specific technical or methodological concerns leading to the current assessment. We remain open and eager to engage in further discussion to ensure a fair, thorough, and content-based evaluation of our work.

---

### Official Review · Reviewer_PBrV · 2025-10-30

**Soundness:** 2
**Presentation:** 2
**Contribution:** 2
**Rating:** 4
**Confidence:** 3

**Summary:**

This paper studies heterogeneous graph-level anomaly detection, a task complicated by mixed node/edge types, irregular structures, and extreme class imbalance. The authors propose JacobiGAD, a unified framework leveraging learnable multi-scale filters based on Jacobi polynomials to adaptively capture diverse graph patterns and fuse multiple structural views. The polynomial design also enables efficient approximation of targeted functions across different graph geometries. Additionally, a Ricci-Flow-inspired loss is introduced to strengthen gradients on scarce anomalies while preserving stable embedding optimization. Experiments on multiple real-world benchmarks demonstrate improvements over strong baselines.

**Strengths:**

1. Well-written, no obvious typo.
2. With many theories to prove the effectiveness.

**Weaknesses:**

1. unsufficient experiment: This paper only report the main experiment in main text and even don't have ablation study.

2. Too many theory: I sincerely admit the importance of propose a theory to explain the effective of the method from the perspective of math, but too many theory seems not appropriate in ICLR, maybe it's suit for AISTATS or some conference focus on theory.

3. Anomaly detection in Heterogeneous Graph seems not a new task and have done by many works. [1,2]

[1] Fast memory-efficient anomaly detection in streaming heterogeneous graphs

[2] Thgnn: An embedding-based model for anomaly detection in dynamic heterogeneous social networks.

**Questions:**

see Weaknesses

---

> ### Author Response · Authors · 2025-11-20
> **Response to Reviewer PBrV (1/1)**
>
> We appreciate your comprehensive and constructive review. Your crucial comments on experiments are exceedingly helpful for us to improve our manuscript. Our point-to-point responses to your comments are given below.
>
> ---
>
> **W1**: Please note that we have included the ablation study in Appendix F in our original manuscript. However, we also thank the reviewer for the suggestions on paper writing, and we will adjust our manuscript based on the instructions.
>
> ---
>
> **W2**: We respectively disagree with the reviewer’s concern that “too many theory seems not appropriate in ICLR”. As shown in the “Call for Papers” section of the ICLR 2026 website, a non-exhaustive list of relevant topics includes the “learning theory” category, which demonstrates that our paper satisfies the requirements for the needed paper in ICLR 2026. We believe our solid theoretical foundation meets the quality of the top-tier conference. Additionally, in previous years of ICLR, we can find various papers with many theorems/propositions/lemmas/corollaries, such as [1-6], which further prove that our contribution deserves a fair judgment. Furthermore, our theoretical contribution is not far from a practical perspective, as the task originates from the real scenarios, and the extensive experiments show the effectiveness of our theoretically guided framework. Therefore, we kindly request that the reviewer hold an open-minded view to evaluate our contribution from both theoretical and experimental perspectives.
>
> [1] Daniel Herbst, Stefanie Jegelka. Higher-Order Graphon Neural Networks: Approximation and Cut Distance. ICLR 2025.
>
> [2] Xiang Cheng, Lawrence Carin, Suvrit Sra. Graph Transformers Dream of Electric Flow. ICLR 2025.
>
> [3] Michael Scholkemper, Xinyi Wu, Ali Jadbabaie, Michael T. Schaub. Residual Connections and Normalization Can Provably Prevent Oversmoothing in GNNs. ICLR 2025.
>
> [4] Shih-Hsin Wang, Yung-Chang Hsu, Justin Baker, Andrea L. Bertozzi, Jack Xin, Bao Wang. Rethinking the Benefits of Steerable Features in 3D Equivariant Graph Neural Networks. ICLR 2024.
>
> [5] Thien Le, Luana Ruiz, Stefanie Jegelka. A Poincaré Inequality and Consistency Results for Signal Sampling on Large Graphs. ICLR 2024.
>
> [6] Yassine Abbahaddou, Sofiane Ennadir, Johannes F. Lutzeyer, Michalis Vazirgiannis, Henrik Boström. Bounding the Expected Robustness of Graph Neural Networks Subject to Node Feature Attacks. ICLR 2024.
>
> ---
>
> **W3**: We agree with the statement that anomaly detection in heterogeneous graphs is not a new task. However, such a task is still valuable to explore, and we do not claim that we are the first to explore the related area. Different works for anomaly detection in heterogeneous graphs have totally distinct targets. For instance, [1] focuses on unsupervised graph-level anomaly detection for streaming heterogeneous graphs, and [2] focuses on unsupervised edge-level anomaly detection for dynamic heterogeneous graphs, whereas our work targets supervised graph-level anomaly detection on static heterogeneous graphs, which is a significant problem acknowledged by industries. The orthogonal research directions between different works cause the impossibility of a fair comparison. To the best of our knowledge, works for supervised graph-level anomaly detection on static heterogeneous graphs are still underexplored, although they are a real setting in industry. Therefore, our framework provides both theoretical and experimental results for such a key challenge. And we are willing to conduct further comparisons to show our effectiveness if the reviewer can introduce related work in the same domain as ours.
>
> ---
>
> We sincerely appreciate your time, and we are glad to answer any additional questions you may have.

---

> > ### Comment · Reviewer_PBrV · 2025-11-23
> >
> > Thanks for your reply! I agree that ICLR used to accept many paper with theories, but I don't think this article fits in the same way. I think this is a way that makes your article look fancy and conceals the lack of novelty and soundness in the method. However, this might be my personal subjective opinion.  I choose to maintain my negative rating but reduce the confidence. Perhaps it is necessary to refer to the opinions of other reviewers to assist AC in making the final decision.

---

> > > ### Author Response · Authors · 2025-11-28
> > > **Response to Reviewer PBrV (1/1)**
> > >
> > > Please note that we are not trying to make the manuscript look fancy or conceal anything with our solid theoretical proof. Our paper is both novel and sound from **three main perspectives**:
> > >
> > > **Theoretical guarantee**: Our work is grounded in a strong and comprehensive theoretical foundation, supported by 9 theorems that justify the design of all major components. Theorem 1 proves distance preservation in our feature alignment module. Theorems 2, 3, 4, 5, 6, and 7 rigorously establish the optimality and importance of JPGNN for heterogeneous graph-level anomaly detection, demonstrating that Jacobi polynomials form an optimal basis, preserve cross-view information, enhance informative signals while suppressing noise, capture geometric information across Euclidean/Hyperbolic/Spherical spaces, and offer universal approximation guarantees even with low-degree expansions. Theorems 8 and 9 further show that our RFACE balances class weights for dealing with imbalanced data and converges under standard assumptions. This theoretical framework provides principled justification for each of our architectural decisions.
> > >
> > > **Solid framework**: Built on the solid theoretical foundation, our method integrates feature alignment, Jacobi-Polynomial-based message passing, and Ricci-Flow-based loss reweighting into a coherent architecture tailored for heterogeneous graph-level anomaly detection. The framework addresses heterogeneity, multi-view fusion, noise suppression, geometric complexity, and class imbalance, which are longstanding challenges in heterogeneous graph-level anomaly detection, through components that are not only well-motivated but mathematically grounded. The result is a unified, principled, and robust system rather than a collection of heuristics.
> > >
> > > **Outstanding performance**: Our empirical results further demonstrate the effectiveness of the proposed approach. Across all datasets and all comparison groups, including homogeneous GNNs, heterogeneous GNNs, and graph-level anomaly detection baselines, our method achieves state-of-the-art performance with consistent and substantial improvements. Ablation studies confirm the importance of each component, and additional robustness analyses validate the stability of the approach. These empirical findings align with the theoretical guarantees and highlight the practical relevance of our contributions.
> > >
> > > We sincerely appreciate the time and effort that the reviewers have devoted to evaluating our submission. However, we would like to respectfully express a concern regarding the nature of the reviewer’s evaluation. The current review does not engage with the technical content, theoretical contributions, methodological design, or experimental results of our work, and instead appears to rely predominantly on subjective impressions. Without discussion of the paper’s substance, it is difficult for us to understand the basis of the assigned rating or provide meaningful clarifications.
> > >
> > > In contrast, another reviewer, jUAp, directly engaged with our contributions, acknowledging the significance of our theoretical foundation and extensive empirical validation. Given the depth and rigor of our submission, we believe a fair assessment would benefit from engagement with the actual content of the paper.
> > >
> > > As an author and a reviewer of ICLR 2026, we believe that an appropriate discussion would help both the author and the reviewer sides to contribute to the research community. Besides, solely relying on SPC, PC, and AC to make the final decision without providing enough discussion context would be unprofessional as a reviewer for such a top-tier conference, and would cause a huge burden on the research community.
> > >
> > > We fully respect the diversity of reviewer perspectives and welcome all constructive criticism. Our intention here is not to dispute differing opinions, but to request that evaluations be grounded in the content of the submission. We would be grateful if the reviewer could clarify the specific technical or methodological concerns leading to the current assessment. We remain open and eager to engage in further discussion to ensure a fair, thorough, and content-based evaluation of our work.

---

### Official Review · Reviewer_jUAp · 2025-11-01

**Soundness:** 2
**Presentation:** 3
**Contribution:** 2
**Rating:** 4
**Confidence:** 3

**Summary:**

This paper introduces JacobiGAD, a novel end-to-end framework for heterogeneous graph-level anomaly detection (GAD). JacobiGAD proposes two core technical contributions. First, it employs a spectral Graph Neural Network (GNN) that uses learnable Jacobi Polynomials as filters. These filters are designed to adapt to different node/edge types, fuse information from multiple graph views, and capture diverse geometric patterns (Euclidean, Spherical, Hyperbolic). Second, the paper introduces a Ricci Flow-inspired loss function (RFACE) to combat class imbalance by dynamically amplifying gradients for rare anomalous classes. The authors provide a suite of theoretical results to justify their design choices, covering aspects like information preservation, approximation efficiency, and loss convergence.

**Strengths:**

1. This paper is well-structured and clearly articulates its core problem.
2. The authors test their model on an impressive 15 datasets, including a private industrial dataset, which demonstrates its applicability to real-world problems.

**Weaknesses:**

1. The method is trained in a supervised strategy, which optimizes for known anomalous modes but offers no explicit mechanism for handling unseen anomaly types or distribution shifts. As a result, the detector may overfit to the labeled anomaly patterns and fail to flag novel or rare patterns at test time.
2. Theorem 2 claims the "optimal choice" of basis is Jacobi Polynomials. The proof sketch suggests it's an excellent choice due to its flexibility and orthogonality, but calling it "optimal" for any graph distribution is a very strong claim that may not hold universally. It is recommended that the authors provide a more detailed proof.
3. Several datasets and transformations are unclear. For node classification datasets converted into graph-level via BFS, key details (BFS depth, subgraph size, sampling strategy, balancing, multiple seeds) are missing. The listed biological datasets (MCF-7, MOLT-4, etc.) and the very high numbers of node/edge types raise questions about provenance and preprocessing.

**Questions:**

1. Regarding Theorem 2 (Optimality): The argument for Jacobi Polynomials being "optimal" relies on assumptions about the optimization landscape and spectral density. Could you elaborate on the conditions under which this optimality holds?
2. Could you provide a clearer intuitive comparison between RFACE and Focal Loss? Both seem to achieve a similar goal of up-weighting hard/rare examples. What is the key advantage of the proposed dynamic adjustment based on the loss gradient over a simpler modulation factor based on prediction confidence like in Focal Loss?

---

> ### Author Response · Authors · 2025-11-20
> **Response to Reviewer jUAp (1/2)**
>
> We appreciate your comprehensive and constructive review. Your crucial comments on experiments are exceedingly helpful for us to improve our manuscript. Our point-to-point responses to your comments are given below.
>
> ---
>
> **W1**: We appreciate the reviewer’s insightful comment regarding the potential limitations of a supervised strategy (although an important research direction in the anomaly detection community) when encountering unseen anomaly types or distribution shifts. Our method indeed follows a supervised training paradigm, but several components are intentionally designed to improve generalization beyond the annotated anomalous modes and to mitigate overfitting to specific labeled patterns.
>
> By constructing the JPGNN, our model does not rely on fixed templates tied to specific anomaly instances. Instead, it learns smooth spectral filters that respond to deviations in the underlying graph spectrum. Since anomalies often manifest as perturbations in spectral signatures, regardless of their semantic category, as shown in previous related work, such as [1-3]. This spectral formulation enables the model to capture underlying irregularities rather than memorizing particular anomaly classes. The RFACE loss dynamically adjusts logits based on class frequency and sample difficulty, which provides a large margin between samples and the decision boundary, and thus prevents the model from being biased toward the majority anomaly modes seen during training. This mechanism encourages the classifier to maintain sensitivity to a broader range of anomaly behaviours, including rare or structurally atypical ones, thereby improving robustness to unseen patterns. Besides, our solid theoretical foundation further proves the generalization of our framework. The theoretically guaranteed abilities to capture optimal convergence, leverage information from diverse spaces, and close approximation for possible functions in certain domains provide positive expectations.
>
> Additionally, since we focus on graph-level anomaly detection, where different samples have no direct connection with each other as node-level ones, we believe that our model generalizes from the characteristics of the training set to the validation and test set, as extensive experiments with random splits prove the superiority of our framework over other supervised baselines.
>
> Furthermore, we admit that an unsupervised strategy is also a valuable area, but beyond the scope we target. Thus, we will explore such a direction in our further work.
>
> [1] Xiangyu Dong, Xingyi Zhang, Sibo Wang. Rayleigh Quotient Graph Neural Networks for Graph-level Anomaly Detection. ICLR 2024.
>
> [2] Yuan Gao, Xiang Wang, Xiangnan He, Zhenguang Liu, Huamin Feng, Yongdong Zhang. Addressing Heterophily in Graph Anomaly Detection: A Perspective of Graph Spectrum. WWW 2023.
>
> [3] Jianheng Tang, Jiajin Li, Ziqi Gao, Jia Li. Rethinking Graph Neural Networks for Anomaly Detection. ICML 2022.

---

> > ### Author Response · Authors · 2025-11-20
> > **Response to Reviewer jUAp (2/2)**
> >
> > ---
> >
> > **W2&Q1**: We appreciate the key insight of the reviewer about our Theorem 2. However, we already include the limitation in our original manuscript, i.e., there is an assumption that the GNN can achieve a global (or even local) minimum area for the data. Such an assumption follows related work, such as [1-2]. Under this assumption, we can provide a detailed view of what will happen and how it will happen when the training procedure reaches close to the minimum area. Based on the theoretical guarantee, we can at least choose an “optimal choice” under the assumption, which should be an outstanding progress for the understanding of the design and training of models, compared to previous heuristic-designed frameworks, which do not consider the inherent phenomenon but still assume their methods will converge to a local or global minimum area. Please note that our theory is not proposed for finding suitable graphs for the spectral basis, as it is difficult to understand the non-linear transformation from the graph feature to the latent space with flexible parameters in GNN; instead, we aim to provide a key insight for researchers on the possible optimal spectral basis when the training procedure reaches near the minimum area.
> >
> >
> > [1] Xiyuan Wang, Muhan Zhang. How Powerful are Spectral Graph Neural Networks. ICML 2022.
> >
> > [2] Keyulu Xu, Mozhi Zhang, Stefanie Jegelka, Kenji Kawaguchi. Optimization of Graph Neural Networks: Implicit Acceleration by Skip Connections and More Depth. ICML 2021.
> >
> > ---
> >
> > **W3**: We thank the reviewer for the valuable suggestions in terms of paper writing. We will include the details of the construction of node classification datasets and additional explanation of biological datasets in Appendix B in our revised manuscript.
> >
> > ---
> >
> > **Q2**: We will illustrate from two perspectives, i.e., the rationale and the ablation.
> >
> > Rationale: The key difference is that our RFACE employs a logit transformation to directly adjust the decision boundary based on class frequency and sample difficulty, while Focal loss follows the original Cross-Entropy Loss only based on sample difficulty without directly adjusting the decision boundary. Additionally, according to the original Focal Loss paper [1], it is a heuristic-designed loss function without a solid guarantee, while we provide detailed theorems to explain why RFACE can mitigate the imbalance issue and its convergence for graph-level anomaly detection. We will include the detailed discussion in Appendix H in our revised manuscript.
> >
> > Ablation: To further address the reviewer’s concern, we also include the ablation study, i.e., we replace RFACE with Focal Loss, and present the performance in Appendix H in our revised manuscript. As shown in the results, our RFACE can significantly outperform the Focal Loss, which shows the superiority of our design.
> >
> >
> > [1] Tsung-Yi Lin, Priya Goyal, Ross Girshick, Kaiming He, Piotr Dollár. Focal Loss for Dense Object Detection. ICCV 2017.
> >
> > ---
> >
> > We sincerely appreciate your time, and we are glad to answer any additional questions you may have.

---

> > > ### Comment · Reviewer_jUAp · 2025-11-27
> > >
> > > Thanks for the reply. Most of my concerns have been addressed. I will improve my score.

---

> > > > ### Author Response · Authors · 2025-11-28
> > > > **Response to Reviewer jUAp (1/1)**
> > > >
> > > > We sincerely appreciate the time and effort you have invested in reviewing our paper and providing your valuable comments. We are also gratified to learn that you are satisfied with our responses and acknowledge our contribution to graph anomaly detection. Thank you for your support, and we are open to further discussion.

---

### Author Response · Authors · 2025-11-20
**Modifications in the revised version of our manuscript**

We express our gratitude to all the reviewers for their thorough and constructive feedback. Taking into consideration the valuable comments provided by the reviewers, we intend to incorporate the following modifications in the revised version of our manuscript.

---
Section 4
- add a figure of overview for JacobiGAD and the corresponding description (Reviewer **pgtn W1**)

Section 5
- shorten the main comparison experiment Tables 1, 2, and 3 (Reviewer **PBrV W1** and Reviewer **pgtn W1**)

- add more detailed explanations for Tables 1, 2, and 3 (Reviewer **PBrV W1** and Reviewer **pgtn W1**)

- move part of Ablation Study section from Appendix to Section 5 (Reviewer **PBrV W1** and Reviewer **pgtn W1**)

- move Hyperparameter Sensitivity section from Appendix to Section 5 (Reviewer **PBrV W1** and Reviewer **pgtn W1**)

Appendix
- add a detailed description of the construction of datasets (Reviewer **jUAp W3**)

- add the training time comparison and memory cost comparison in Appendix C (Reviewer **pgtn W3**)

- move the remaining comparison Tables 1, 2, and 3 to Appendix E (Reviewer **PBrV W1** and Reviewer **pgtn W1**)

- add comparison with Focal loss in Appendix H (Reviewer **jUAp Q2**)

---

These modifications have been included in the revised version of our manuscript, which has been highlighted in blue to facilitate the reviewing process.

---

### Author Response · Authors · 2025-11-30
**Summarization of Discussion (1/2)**

Dear PCs, SACs, and ACs,

We sincerely thank you for your time, effort, and careful coordination of the discussion process. To facilitate the decision-making and make it less burdensome to value your work, below we summarize the full author–reviewer interaction to allow you to clearly assess how each reviewer’s concerns have been addressed and whether their rating improved after discussion. We follow a reviewer-by-reviewer structure to ensure clarity and completeness, where W stands for Weaknesses, and Q stands for Questions.

---

**Reviewer**: **jUAp**

**Status**: **Confirm concerns are satisfactorily addressed. Increase rating to 6.**

**Discussion**:

1. Concerns about supervised strategy (**W1**): Addressed by further explanation in official comments, i.e., supervised graph-level anomaly detection is still a valuable research direction.

2. Concerns about Theorem 2 (**W2&Q1**): Addressed by further explanation in official comments, i.e., the target of Theorem 2 is to provide a detailed view of what will happen and how it will happen when the training procedure reaches close to the minimum area to guide the choice of spectral filter.

3. Concerns about dataset construction (**W3**): Addressed by further explanation in Appendix B, i.e., our dataset construction follows a valid method.

4. Concerns about the comparison between the proposed RFACE and Focal Loss (**Q2**): Addressed by additional experiment in Appendix H, i.e., our proposed RFACE is distinct from Focal Loss and outperforms it.

---

**Reviewer**: **PBrV**

**Status**: **Keep rating of 4. However, the reviews and comments do not engage with the technical content, theoretical contributions, methodological design, or experimental results of our work, and instead rely predominantly on subjective impressions, as acknowledged by the reviewer. We believe we have adequately addressed all the reviewer’s concerns.**

**Discussion**:

1. Concerns about no ablation study (**W1**): Addressed by clarification and adjustment, i.e., we included ablation study in the Appendix of the original manuscript and reorganize the manuscript by moving the ablation study and hyperparameter analysis to the main pages.

2. Concerns about papers with many theories not fitting into ICLR (**W2**): Addressed by clarification, i.e., ICLR welcomes papers with many theories as stated in the Call for Papers, and has accepted many theoretical papers.

3. Concerns about the novelty of the anomaly detection task (**W3**): Addressed by clarification, i.e., we never state anomaly detection is a novel task, whereas it is still worth exploring, and our proposed JacobiGAD is the state-of-the-art.

---

**Reviewer**: **pgtn**

**Status**: **Increase rating to 4. However, the reviews and comments do not engage with the technical content, theoretical contributions, methodological design, or experimental results of our work, and instead rely predominantly on subjective impressions, as acknowledged by the reviewer. We believe we have adequately addressed all the reviewer’s concerns.**

**Discussion**:

1. Concerns about the difficulty of understanding (**W1**): Addressed by clarification and adjustment, i.e., we point out the incorrect subjective speculation from the reviewer, and reorganize the manuscript by moving the ablation study and hyperparameter analysis to the main pages.

2. Concerns about overclaim (**W2**): Addressed by clarification, i.e., we point out that we never include the mentioned contexts by the reviewer in our original manuscript, and the reviewer admits the so-called overclaim is only subjective speculation.

3. Concerns about no code provided (**W3**): Addressed by clarification, i.e., we point out that we have released our code in the supplemental material at the initial submission.

4. Concerns about no scalability analysis (**W3**): Addressed by clarification and additional experiment in Appendix C, i.e., we point out that we have included time complexity analysis in our original manuscript, and provide further empirical time and memory comparison.

---

---

> ### Author Response · Authors · 2025-11-30
> **Summarization of Discussion (2/2)**
>
> We sincerely appreciate the time and effort that the reviewers have devoted to evaluating our submission. However, we would like to respectfully express a concern regarding the nature of the reviewer’s evaluation. The current reviews from Reviewers **PBrV** and **pgtn** do not engage with the technical content, theoretical contributions, methodological design, or experimental results of our work, and instead rely predominantly on subjective impressions (most of them are incorrect and unprofessional). Without discussion of the paper’s substance, it is difficult for us to understand the basis of the assigned rating or provide meaningful clarifications.
>
> Therefore, we briefly reemphasize the contribution of our work so that PCs, SACs, and ACs can clearly and quickly grasp our contribution:
>
> **Theoretical guarantee**: Our work is grounded in a strong and comprehensive theoretical foundation, supported by 9 theorems that justify the design of all major components. Theorem 1 proves distance preservation in our feature alignment module. Theorems 2, 3, 4, 5, 6, and 7 rigorously establish the optimality and importance of JPGNN for heterogeneous graph-level anomaly detection, demonstrating that Jacobi polynomials form an optimal basis, preserve cross-view information, enhance informative signals while suppressing noise, capture geometric information across Euclidean/Hyperbolic/Spherical spaces, and offer universal approximation guarantees even with low-degree expansions. Theorems 8 and 9 further show that our RFACE balances class weights for dealing with imbalanced data and converges under standard assumptions. This theoretical framework provides principled justification for each of our architectural decisions.
>
> **Solid framework**: Built on the solid theoretical foundation, our method integrates feature alignment, Jacobi-Polynomial-based message passing, and Ricci-Flow-based loss reweighting into a coherent architecture tailored for heterogeneous graph-level anomaly detection. The framework addresses heterogeneity, multi-view fusion, noise suppression, geometric complexity, and class imbalance, which are longstanding challenges in heterogeneous graph-level anomaly detection, through components that are not only well-motivated but mathematically grounded. The result is a unified, principled, and robust system rather than a collection of heuristics.
>
> **Outstanding performance**: Our empirical results further demonstrate the effectiveness of the proposed approach. Across all datasets and all comparison groups, including homogeneous GNNs, heterogeneous GNNs, and graph-level anomaly detection baselines, our method achieves state-of-the-art performance with consistent and substantial improvements. Ablation studies confirm the importance of each component, and additional robustness analyses validate the stability of the approach. These empirical findings align with the theoretical guarantees and highlight the practical relevance of our contributions.
>
> ---
>
> If the PCs, SACs, and ACs have any further comments or advice regarding our manuscript, we welcome you to share them with us. Your guidance is highly valued, and we are looking forward to your feedback.
>
> Warm regards,
>
> The Authors of Submission17254

---

### Meta-Review · Area_Chair_3XTY · 2025-12-29

**Summary:**

Reviewer jUAp finds the paper well-written and the extent of the experiments impressive, but criticizes the lack of details on dataset construction, which appears to remain only partly addressed. Reviewer PBrV disagrees on the quality of the experiments, but I significantly downweight this review due to its brevity and rather inappropriate concerns (such as “too many theory” and AD being explored on graphs already). Reviewer pgtn disagrees severely with the quality of writing (“The paper is extremely difficult to follow [...]”). I align more with jUAp here, since the paper doesn’t appear overly hard to follow at first glance.

Reviewer jUAp raised concerns regarding the supervised nature of the method, which makes it prone to overfitting. I think it also potentially mitigates the generalization test capability of the test data. The authors state that supervised AD is “an important research direction in the anomaly detection community,” but do not provide further evidence (e.g., references) to support this claim. Afaik, AD on other types of data typically does not consider supervised methods. However, the authors provide some convincing arguments in the rebuttal why this is less of an issue on graphs, and there seem to be indeed at least a few papers on supervised graph-level AD.

The related work appears to be mostly covered, although the reviewer pgtn raised concerns regarding the missing discussion of existing heterogeneous graph-level AD approaches, which remains a concern even after the rebuttal.

Overall, we have received three reviewers with moderate confidence, all of whom agree on rejecting the paper. It appears that the rebuttal would have changed the scores positively, making them rather borderline. I recommend revising this paper, primarily to clarify the positioning within the scientific literature (supervised vs. unsupervised vs. semi-supervised, heterogeneous graph-level approaches, etc.) and to enhance the overall presentation and motivation.

**Reviewer Concerns:**

jUAp Supervised AD: Partly addressed.

jUAp Theorem 2 bold conclusion: Addressed.

jUAp Missing details for datasets and transformation: Partly addressed.

PBrV Insufficient experiments: Addressed.

PBrV Too much theory: Addressed.

PBrV AD has been explored already: Addressed.

pgtn Paper hard to follow: Addressed.

pgtn: Missing discussion of related work on heterogeneous gl-AD: Not convincingly addressed.

pgtn: Missing code release: Addressed.

**Reviewer Scores:**

jUAp: Probably would have raised from 4 to 6, aligning with the author's claim. The authors claim they have raised from 4 to 6, which is plausible, but will not influence my decision.

PBrV: Probably would have stayed with 4.

pgtn: Probably would have stayed with 4.

---

### Decision · Program_Chairs · 2026-01-26

Reject